# Rethinking Consistent Multi-Label Classification Under Inexact Supervision

**Wei Wang** [1,2*†]    **Tianhao Ma** [2*]    **Ming-Kun Xie** [1]    **Gang Niu** [1]    **Masashi Sugiyama** [1,2]

[1] RIKEN, Japan
[2] The University of Tokyo, Japan
`{wei.wang,ming-kun.xie}@riken.jp`
`{matianhao2120,gang.niu.ml}@gmail.com sugi@k.u-tokyo.ac.jp`

## Abstract

Partial multi-label learning and complementary multi-label learning are two popular weakly supervised multi-label classification paradigms that aim to alleviate the high annotation costs of collecting precisely annotated multi-label data. In partial multi-label learning, each instance is annotated with a candidate label set, among which only some labels are relevant; in complementary multi-label learning, each instance is annotated with complementary labels indicating the classes to which the instance does not belong. Existing consistent approaches for the two paradigms either require accurate estimation of the generation process of candidate or complementary labels or assume a uniform distribution to eliminate the estimation problem. However, both conditions are usually difficult to satisfy in real-world scenarios. In this paper, we propose consistent approaches that do not rely on the aforementioned conditions to handle both problems in a unified way. Specifically, we propose two risk estimators based on first- and second-order strategies. Theoretically, we prove consistency w.r.t. two widely used multi-label classification evaluation metrics and derive convergence rates for the estimation errors of the proposed risk estimators. Empirically, extensive experimental results on both real-world and synthetic datasets validate the effectiveness of our proposed approaches against state-of-the-art methods.

## 1 Introduction

In multi-label classification (MLC), each instance is associated with multiple relevant labels simultaneously (Zhang & Zhou, 2014; Liu et al., 2022b). The goal of MLC is to induce a multi-label classifier that can assign multiple relevant labels to unseen instances. MLC is more practical and useful than single-label classification, as real-world objects often appear together in a single scene. The ability to handle complex semantic information has led to the widespread use of MLC in many real-world applications, including multimedia content annotation (Cabral et al., 2011), text classification (Rubin et al., 2012; Liu et al., 2017), and music emotion analysis (Wu et al., 2014). However, annotating multi-label training data is more expensive and demanding than annotating single-label data. This is because each instance can be associated with an unknown number of relevant labels (Durand et al., 2019; Cole et al., 2021; Xie et al., 2023), making it difficult to collect a large-scale multi-label dataset with precise annotations.

To address this, learning from weak supervision has become a prevailing way to mitigate the bottleneck of annotation cost for MLC (Sugiyama et al., 2022). Among them, partial multi-label learning (PML) and complementary multi-label learning (CML) have become two popular MLC paradigms. In PML, each instance is annotated with a *candidate label set*, among which only some labels are relevant but inaccessible to the learning algorithm (Xie & Huang, 2018; Sun et al., 2019; Gong et al., 2021). In CML, each instance is annotated with *complementary labels*, which indicate the classes to which the instance does not belong (Gao et al., 2023; Zhu et al., 2025; Gao et al., 2026). Given that all relevant labels are included in the candidate label set, non-candidate labels

---

*Equal contributions. [†]Corresponding author: Wei Wang <wei.wang@riken.jp>.

Table 1: Comparison of COMES with existing consistent PML and CML approaches.

| Approach | Uniform distribution assumption-free | Generation process estimation unnecessary | Label correlation-aware | Multiple complementary labels |
|---|---|---|---|---|
| CCMN (Xie & Huang, 2023) | ✓ | ✗ | ✓ | ✓ |
| CTL (Gao et al., 2023) | ✗ | ✓ | ✗ | ✗ |
| MLCL (Gao et al., 2024) | ✓ | ✗ | ✓ | ✗ |
| GDF (Gao et al., 2025) | ✗ | ✓ | ✗ | ✓ |
| COMES-HL (Ours) | ✓ | ✓ | ✗ | ✓ |
| COMES-RL (Ours) | ✓ | ✓ | ✓ | ✓ |

contain no relevant labels and can be considered complementary labels, and vice versa. This suggests that the two problems are mathematically equivalent. Therefore, in this paper, we treat them as *MLC under inexact supervision* in a unified way. Figure 1 shows an example image annotated with inexact annotations. The label space contains ten labels in total. The candidate label set consists of four relevant labels {*apple*, *plate*, *table*, *jug*} and two false-positive ones {*grapes*, *pear*}. By excluding the candidate labels from the label space, the remaining four labels are {*banana*, *cup*, *knife*, *flower*}, which can be considered complementary labels. PML and CML do not require precise determination of all relevant labels during annotation, which demonstrates their great potential for alleviating annotation challenges in MLC.

In this paper, we investigate *consistent* approaches for MLC under inexact supervision. Here, consistency means that classifiers learned with inexact supervision are theoretically guaranteed to converge to the optimal classifiers when infinitely many training samples are provided (Wang et al., 2024). The remedy began with Xie & Huang (2023), which treated PML as a special case of MLC with class-conditional label noise (Li et al., 2022; Xia et al., 2023), where irrelevant labels could flip to relevant labels but not vice versa. However, the flipping

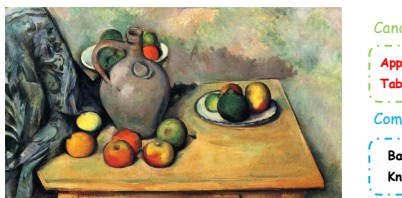

Figure 1: A multi-label image with inexact annotations. Source: Paul Cézanne, Still Life, Jug and Fruit on a Table (1894), public domain.

rate for each class is unknown and must be estimated using anchor points, i.e., instances belonging to a specific class with probability one (Liu & Tao, 2015; Xie & Huang, 2023). Similar to PML, CML assumes that complementary labels are generated by a certain flipping process (Yu et al., 2018b). Gao et al. (2023) proposed the *uniform distribution assumption* that a label outside the relevant label set is sampled uniformly to be the CL. Then, Gao et al. (2024) generalized the data generation process with a transition matrix, but estimating the data generation process is still necessary. Recently, Gao et al. (2025) extended the uniform distribution assumption to handle multiple complementary labels.

In summary, all existing consistent PML and CML approaches either estimate the generation process of the candidate label set or complementary labels, or adopt the uniform distribution assumption to eliminate the estimation problem. However, both conditions are difficult to satisfy in real-world scenarios. On the one hand, estimating the flipping rate heavily relies on accurate estimation of noisy class posterior probabilities of anchor points (Xia et al., 2019; Yao et al., 2020; Lin et al., 2023). However, estimating noisy class posterior probabilities is more difficult because their entropy is usually higher than that of clean labels (Langford, 2005). This difficulty is further amplified when using deep neural networks, where the over-confidence phenomenon typically occurs (Zhang et al., 2021; Wei et al., 2022). The model outputs of deep neural networks are usually one-hot encoded, which means they cannot yield reliable probabilistic outputs (Guo et al., 2017). On the other hand, the uniform distribution assumption treats different candidate label sets indiscriminately, which is too simple to be truly in accordance with imbalanced classes in real-world scenarios (Wang et al., 2025). Additionally, many approaches model different labels independently and directly ignore label correlations existing in multi-label data (Gao et al., 2025). This prevents them from exploiting the rich semantic relationships of label correlations (Zhu et al., 2017; Mao et al., 2023).

To this end, we propose a novel framework named COMES, i.e., *COnsistent Multi-label classification under inExact Supervision*. Based on a data generation process that does not use transition ma-

trices, we introduce two instantiations with risk estimators w.r.t. the Hamming loss and ranking loss, respectively. Table 1 compares our approach with existing consistent PML and CML approaches. Our contributions are summarized as follows:

- We propose a consistent framework for multi-label classification under inexact supervision that neither requires estimating the generation process of candidate or complementary labels nor relies on the uniform-distribution assumption.
- We introduce risk-correction approaches to improve the generalization performance of the proposed risk estimators. We further prove consistency w.r.t. two widely used metrics and derive convergence rates of estimation errors for the proposed risk estimators.
- Our proposed approaches outperform state-of-the-art baselines on both real-world and synthetic PML and CML datasets with different label generation processes.

## 2 PRELIMINARIES

In this section, we introduce the background of MLC and MLC under inexact supervision.

### 2.1 MULTI-LABEL CLASSIFICATION

Let $\mathcal{X} \subseteq \mathbb{R}^d$ denote the $d$-dimensional feature space and $\mathcal{Y} = \{1, 2, \ldots, q\}$ the label space consisting of $q$ class labels. A multi-label example is denoted as $(\boldsymbol{x}, Y)$, where $\boldsymbol{x} \in \mathcal{X}$ is a feature vector and $Y \subseteq \mathcal{Y}$ is the set of relevant labels associated with $\boldsymbol{x}$. For ease of notation, we introduce $\boldsymbol{y} = [y_1, y_2, \ldots, y_q] \in \{0, 1\}^q$ to denote the vector representation of $Y$, where $y_j = 1$ if $j \in Y$ and $y_j = 0$ otherwise. Let $p(\boldsymbol{x}, Y)$ denote the joint density of $\boldsymbol{x}$ and $Y$. Let $p(\boldsymbol{x})$ denote the marginal density, and $\pi_j = p(y_j = 1)$ the prior of the $j$-th class. The task of MLC is to learn a prediction function $\boldsymbol{f} : \mathcal{X} \mapsto 2^{\mathcal{Y}}$. We use $f_j$ to denote the $j$-th entry of $\boldsymbol{f}$, where $f_j(\boldsymbol{x}) = 1$ indicates that the model predicts class $j$ to be relevant to $\boldsymbol{x}$ and $f_j(\boldsymbol{x}) = 0$ otherwise. Since learning $\boldsymbol{f}$ directly is often difficult, we use a real-valued decision function $\boldsymbol{g} : \mathcal{X} \mapsto \mathbb{R}^q$ to represent the model output. The prediction function $\boldsymbol{f}$ can be derived by thresholding $\boldsymbol{g}$. We use $g_j$ to denote the $j$-th entry of $\boldsymbol{g}$, which indicates the model output for class $j$.

Many evaluation metrics have been developed to calculate the difference between model predictions and true labels to evaluate the performance of multi-label classifiers (Zhang & Zhou, 2014; Wu & Zhou, 2017). In this paper, we focus primarily on the Hamming loss and ranking loss, the two most common metrics in the literature.[1] Specifically, the Hamming loss calculates the fraction of misclassified instance-label pairs, and the risk of $\boldsymbol{f}$ w.r.t. the Hamming loss is

$$R_{\mathrm{H}}^{0-1}(\boldsymbol{f}) = \mathbb{E}_{p(\boldsymbol{x}, Y)} \left[ \frac{1}{q} \sum_{j=1}^{q} \mathbb{I}\left(f_j(\boldsymbol{x}) \neq y_j\right) \right]. \tag{1}$$

Here, $\mathbb{I}$ denotes the indicator function that returns 1 if the predicate holds; otherwise, $\mathbb{I}$ returns 0. Since optimizing the 0-1 loss is difficult, a surrogate loss function $\ell$ is often adopted. The $\ell$-risk w.r.t. the Hamming loss is

$$R_{\mathrm{H}}^{\ell}(\boldsymbol{g}) = \mathbb{E}_{p(\boldsymbol{x}, Y)} \left[ \frac{1}{q} \sum_{j=1}^{q} \ell\left(g_j\left(\boldsymbol{x}\right), y_j\right) \right], \tag{2}$$

where $\ell$ is a non-negative binary loss function, such as the binary cross-entropy loss. It is important to note that the Hamming loss only considers first-order model predictions and cannot account for label correlations. The ranking loss explicitly considers the ordering relationship between model outputs for a pair of labels. Specifically, the risk of $\boldsymbol{f}$ w.r.t. the ranking loss is[2]

$$R_{\mathrm{R}}^{0-1}(\boldsymbol{f}) = \mathbb{E}_{p(\boldsymbol{x}, Y)} \left[ \sum_{1 \leqslant j < k \leqslant q} \mathbb{I}(y_j < y_k) \left( \mathbb{I}\left(f_j(\boldsymbol{x}) > f_k(\boldsymbol{x})\right) + \frac{1}{2} \mathbb{I}\left(f_j(\boldsymbol{x}) = f_k(\boldsymbol{x})\right) \right) \right.$$
$$\left. + \mathbb{I}(y_j > y_k) \left( \mathbb{I}\left(f_j(\boldsymbol{x}) < f_k(\boldsymbol{x})\right) + \frac{1}{2} \mathbb{I}\left(f_j(\boldsymbol{x}) = f_k(\boldsymbol{x})\right) \right) \right]. \tag{3}$$

---

[1]We will address the use of other metrics in future work.

[2]To facilitate the analysis in this paper, we consider the coefficients of the losses for different label pairs to be 1 (Gao & Zhou, 2013; Xie & Huang, 2021; 2023).

Similarly, when using a surrogate loss function $\ell$ to replace the 0-1 loss, the $\ell$-risk w.r.t. the ranking loss is

$$R_{\mathrm{R}}^{\ell}(\boldsymbol{g}) = \mathbb{E}_{p(\boldsymbol{x},Y)}\left[\sum_{1 \leqslant j < k \leqslant q} \mathbb{I}(y_j \neq y_k)\ell\left(g_j(\boldsymbol{x}) - g_k(\boldsymbol{x}), \frac{y_j - y_k + 1}{2}\right)\right]. \tag{4}$$

Notably, minimizing the Hamming loss does not consider label correlations and can be considered a first-order strategy. In contrast, minimizing the ranking loss considers label-ranking relationships and can be considered a second-order strategy.

## 2.2 MULTI-LABEL CLASSIFICATION UNDER INEXACT SUPERVISION

In PML, each example is denoted as $(\boldsymbol{x}, S)$, where $S$ is the candidate label set associated with $\boldsymbol{x}$. The basic assumption of PML is that all relevant labels are contained within the candidate label set, i.e., $Y \subseteq S$. Let $\bar{S} = \mathcal{Y}\backslash S$ denote the *absolute complement* of $S$. Since $\bar{S} \bigcap Y = \varnothing$, $\bar{S}$ can be regarded as the set of complementary labels associated with $\boldsymbol{x}$. Therefore, PML and CML are mathematically equivalent, as partial multi-label data can equivalently be transformed into complementary multi-label data and vice versa. Without loss of generality, this paper mainly considers partial multi-label data. For ease of notation, we use $\boldsymbol{s} = [s_1, s_2, \ldots, s_q]$ to denote the vector representation of $S$. Here, $s_j = 1$ indicates that the $j$-th class label is a candidate label of $\boldsymbol{x}$, and $s_j = 0$ otherwise. Let $p(\boldsymbol{x}, S)$ denote the joint density of $\boldsymbol{x}$ and the candidate label set $S$. The goal of PML or CML is to learn a prediction function $\boldsymbol{f} : \mathcal{X} \mapsto 2^{\mathcal{Y}}$ that can assign relevant labels to unseen instances based on a training set $\mathcal{D} = \{(\boldsymbol{x}_i, S_i)\}_{i=1}^{n}$ sampled i.i.d. from $p(\boldsymbol{x}, S)$.

## 3 METHODOLOGY

In this section, we first introduce our data generation process. Then, we present the first- and second-order strategies for handling the PML problem and their respective theoretical analyses.

## 3.1 DATA GENERATION PROCESS

In this paper, we assume that the candidate labels are generated by querying *whether each instance is irrelevant to a class* in turn. Specifically, if the $j$-th class is irrelevant to $\boldsymbol{x}$, we assume that the $j$-th class label is assigned as a non-candidate label to $\boldsymbol{x}$ with a constant probability $p_j$, i.e., $p(j \notin S|\boldsymbol{x}, j \notin Y) = p_j$. Otherwise, if the $j$-th class is relevant to $\boldsymbol{x}$, we consider it as a candidate label. The candidate label set can then be obtained by excluding the non-candidate labels from the label space. Notably, all relevant labels are included in the candidate label set, as well as some irrelevant labels. This data generation process coincides well with the annotation process of candidate labels. For example, when asking annotators to provide candidate labels for an image dataset, we can show them an image and a class label and ask them to determine whether the image is irrelevant to that class. This is often an easier question to answer than directly asking all relevant labels, since it is less demanding to exclude some obviously irrelevant labels. If so, we assume that the image will be annotated with this label as a non-candidate label with a constant probability. Based on this data generation process, we have the following lemma.

**Lemma 1.** *Assume that $p(s_j = 0|\boldsymbol{x}, y_j = 0) = p_j$, where $p_j$ is a constant. Then, we have $p(\boldsymbol{x}|s_j = 0) = p(\boldsymbol{x}|y_j = 0)$.*

The proof can be found in Appendix B.1. According to Lemma 1, the conditional density of instances where the $j$-th class is considered a non-candidate label is equivalent to the conditional density of instances where the $j$-th class is irrelevant. Notably, our data distribution assumption differs from both the uniform distribution assumption and the use of a transition matrix to flip the labels. Since the conditional probabilities of different candidate label sets can be different, our setting is more general than the uniform distribution assumption (Gao et al., 2023; 2025).

## 3.2 FIRST-ORDER STRATEGY

A common strategy used in MLC is to decompose the problem into a number of binary classification problems by ignoring label correlations. This goal can be achieved by minimizing the $\ell$-risk w.r.t. the

Hamming loss in Eq. (2). We show that the $\ell$-risk w.r.t. the Hamming loss can be equivalently expressed with partial multi-label data.

**Theorem 1.** *By the assumption in Lemma 1, the $\ell$-risk w.r.t. the Hamming loss in Eq. (2) can be equivalently expressed as*

$$R_{\mathrm{H}}^{\ell}(\boldsymbol{g}) = \mathbb{E}_{p(\boldsymbol{x})}\left[\frac{1}{q}\sum_{j=1}^{q}\ell\left(g_j\left(\boldsymbol{x}\right),1\right)\right]$$
$$+ \sum_{j=1}^{q}\mathbb{E}_{p(\boldsymbol{x}|s_j=0)}\left[\frac{1-\pi_j}{q}\left(\ell\left(g_j\left(\boldsymbol{x}\right),0\right)-\ell\left(g_j\left(\boldsymbol{x}\right),1\right)\right)\right]. \tag{5}$$

The proof can be found in Appendix B.2. Theorem 1 shows that the $\ell$-risk w.r.t. the Hamming loss can be expressed as the expectation w.r.t. the marginal and conditional densities where the $j$-th class label is not considered as a candidate label. Since Eq. (5) cannot be calculated directly, we perform *empirical risk minimization (ERM)* by approximating Eq. (5) using datasets $\mathcal{D}_{\mathrm{U}}$ and $\mathcal{D}_j (j \in \mathcal{Y})$ sampled from densities $p(\boldsymbol{x})$ and $p(\boldsymbol{x}|s_j = 0)$, respectively. In this paper, we consider generating these datasets by *duplicating* instances from $\mathcal{D}$. Specifically, we first treat the duplicated instances of $\mathcal{D}$ as unlabeled data sampled from $p(\boldsymbol{x})$ and add them to $\mathcal{D}_{\mathrm{U}}$. Then, if an instance does not treat the $j$-th class label as a candidate label, we treat its duplicated instance as being sampled from $p(\boldsymbol{x}|s_j = 0)$ and add it to $\mathcal{D}_j$. These processes can be expressed as follows:

$$\mathcal{D}_{\mathrm{U}} = \left\{\boldsymbol{x}_i^{\mathrm{U}}\right\}_{i=1}^{n} = \left\{\boldsymbol{x}_i | (\boldsymbol{x}_i, S_i) \in \mathcal{D}\right\}, \quad \mathcal{D}_j = \left\{\boldsymbol{x}_i^{j}\right\}_{i=1}^{n_j} = \left\{\boldsymbol{x}_i | (\boldsymbol{x}_i, S_i) \in \mathcal{D}, j \notin S_i\right\}, j \in \mathcal{Y}. \tag{6}$$

Then, an unbiased risk estimator can be derived to approximate Eq. (5) using datasets $\mathcal{D}_{\mathrm{U}}$ and $\mathcal{D}_j$:

$$\hat{R}_{\mathrm{H}}^{\ell}(\boldsymbol{g}) = \frac{1}{nq}\sum_{i=1}^{n}\sum_{j=1}^{q}\ell\left(g_j\left(\boldsymbol{x}_i^{\mathrm{U}}\right),1\right)$$
$$+ \sum_{j=1}^{q}\frac{1-\pi_j}{qn_j}\sum_{i=1}^{n_j}\left(\ell\left(g_j\left(\boldsymbol{x}_i^{j}\right),0\right)-\ell\left(g_j\left(\boldsymbol{x}_i^{j}\right),1\right)\right). \tag{7}$$

When deep neural networks are used, the negative terms in the loss function can often lead to overfitting issues (Kiryo et al., 2017; Sugiyama et al., 2022). Therefore, we use an absolute value function to wrap each potentially negative term Lu et al. (2020); Wang et al. (2023). The *corrected risk estimator* is defined as

$$\tilde{R}_{\mathrm{H}}^{\ell}(\boldsymbol{g}) = \frac{1}{q}\sum_{j=1}^{q}\left|\frac{1}{n}\sum_{i=1}^{n}\ell\left(g_j\left(\boldsymbol{x}_i^{\mathrm{U}}\right),1\right)-\frac{1-\pi_j}{n_j}\sum_{i=1}^{n_j}\ell\left(g_j\left(\boldsymbol{x}_i^{j}\right),1\right)\right|$$
$$+ \sum_{j=1}^{q}\frac{1-\pi_j}{qn_j}\sum_{i=1}^{n_j}\ell\left(g_j\left(\boldsymbol{x}_i^{j}\right),0\right). \tag{8}$$

Notably, our framework is very flexible so that the minimizer can be obtained using any network architecture and stochastic optimizer. The algorithmic details are summarized in Algorithm 1. The class prior $\pi_j$ can be estimated by using off-the-shelf class prior estimation approaches only using candidate labels (see Appendix A.2).

We establish the consistency and estimation error bounds for the risk estimator proposed in Eq. (8). First, we demonstrate that the corrected risk estimator in Eq. (8) is biased yet consistent w.r.t. the $\ell$-risk w.r.t. the Hamming loss in Eq. (2). The following theorem holds.

**Theorem 2.** *Assume that there exists a constant $C_{\mathcal{G}}$ such that $\sup_{g_j \in \mathcal{G}} \|g_j\|_{\infty} \leqslant C_{\mathcal{G}}$ and a constant $C_{\ell}$ such that $\sup_{|z| \leqslant C_{\mathcal{G}}} \ell(z, y) \leqslant C_{\ell}$, where $\mathcal{G}$ is the model class. We assume that there exists a positive constant $\alpha$ such that $\forall j \in \mathcal{Y}, \pi_j \mathbb{E}_{p(\boldsymbol{x}|y_j=1)}\left[\ell\left(g_j\left(\boldsymbol{x}\right),1\right)\right] \geqslant \alpha$. Then, the bias of the expectation of the corrected risk estimator w.r.t. the $\ell$-risk w.r.t. the Hamming loss has the following lower and upper bounds:*

$$0 \leqslant \mathbb{E}\left[\tilde{R}_{\mathrm{H}}^{\ell}(\boldsymbol{g})\right] - R_{\mathrm{H}}^{\ell}(\boldsymbol{g}) \leqslant \frac{1}{q}\sum_{j=1}^{q}(4-2\pi_j)C_{\ell}\Delta_j, \tag{9}$$

*where $\Delta_j = \exp\left(-2\alpha^2/\left(C_{\ell}^2/n + (1-\pi_j)^2 C_{\ell}^2/n_j\right)\right)$. Furthermore, for any $\delta > 0$, the following inequality holds with probability at least $1 - \delta$:*

$$\left|\tilde{R}_{\mathrm{H}}^{\ell}(\boldsymbol{g}) - R_{\mathrm{H}}^{\ell}(\boldsymbol{g})\right| \leqslant \frac{1}{q}\sum_{j=1}^{q}\left((4-2\pi_j)C_{\ell}\Delta_j + \frac{(2-2\pi_j)C_{\ell}}{q}\sqrt{\frac{\ln\left(2/\delta\right)}{2n_j}}\right) + C_{\ell}\sqrt{\frac{\ln\left(2/\delta\right)}{2n}}. \tag{10}$$

The proof can be found in Appendix B.3. Notably, the bias of the corrected risk estimator from the original $\ell$-risk exists since it is lower bounded by zero. However, as $n \to \infty$, we have that $\tilde{R}_{\mathrm{H}}^{\ell}(\boldsymbol{g}) \to R_{\mathrm{H}}^{\ell}(\boldsymbol{g})$, meaning that it is still consistent.

Let $\tilde{\boldsymbol{g}}_{\mathrm{H}} = \arg\min_{\{g_j\} \subseteq \mathcal{G}} \tilde{R}_{\mathrm{H}}^{\ell}(\boldsymbol{g})$ and $\boldsymbol{g}_{\mathrm{H}}^* = \arg\min_{\{g_j\} \subseteq \mathcal{G}} R_{\mathrm{H}}^{\ell}(\boldsymbol{g})$ denote the minimizer of the corrected risk estimator and the $\ell$-risk w.r.t. the Hamming loss, respectively. Let $\mathfrak{R}_{n,p}(\mathcal{G})$ and $\mathfrak{R}_{n_j,p_j}(\mathcal{G})$ denote the Rademacher complexities defined in Appendix B.4.

**Theorem 3.** *Assume that the loss function $\ell(z, y)$ is Lipschitz continuous in $z$ with a Lipschitz constant $L_\ell$. By the assumptions in Theorem 2, for any $\delta > 0$, the following inequality holds with probability at least $1 - \delta$:*

$$R_{\mathrm{H}}^{\ell}(\tilde{\boldsymbol{g}}_{\mathrm{H}}) - R_{\mathrm{H}}^{\ell}(\boldsymbol{g}_{\mathrm{H}}^*) \leqslant \frac{8L_\ell}{q} \sum_{j=1}^{q} \mathfrak{R}_{n,p}(\mathcal{G}) + \frac{16(1 - \pi_j)L_\ell}{q} \sum_{j=1}^{q} \mathfrak{R}_{n_j,p_j}(\mathcal{G})$$

$$+ \frac{2}{q} \sum_{j=1}^{q} (4 - 2\pi_j)C_\ell \Delta_j + 2C_\ell \sqrt{\frac{\ln(1/\delta)}{2n}} + \sum_{j=1}^{q} \frac{(4 - 4\pi_j)C_\ell}{q} \sqrt{\frac{\ln(1/\delta)}{2n_j}}. \quad (11)$$

The proof can be found in Appendix B.4. Theorem 3 shows that, as $n \to \infty$, $R_{\mathrm{H}}^{\ell}(\tilde{\boldsymbol{g}}_{\mathrm{H}}) \to R_{\mathrm{H}}^{\ell}(\boldsymbol{g}_{\mathrm{H}}^*)$, since $\Delta_j \to 0$, $\mathfrak{R}_{n,p}(\mathcal{G}) \to 0$, and $\mathfrak{R}_{n_j,p_j}(\mathcal{G}) \to 0$ for all parametric models with a bounded norm (Mohri et al., 2012). This means that the minimizer of the corrected risk estimator will approach the desired classifier that minimize the $\ell$-risk w.r.t. the Hamming loss.

Let $R_{\mathrm{H}}^{\ell*} = \inf_{\boldsymbol{g}} R_{\mathrm{H}}^{\ell}(\boldsymbol{g})$ and $R_{\mathrm{H}}^* = \inf_{\boldsymbol{f}} R_{\mathrm{H}}^{0-1}(\boldsymbol{f})$ denote the minima of the $\ell$-risk and the risk w.r.t. the Hamming loss, respectively. Then, the following corollary holds.

**Corollary 1.** *If $\ell$ is a convex function such that $\forall y, \ell'(0, y) < 0$, then the $\ell$-risk w.r.t. the Hamming loss in Eq. (5) is consistent with the risk w.r.t. the Hamming loss in Eq. (1). This means that, for any sequence of decision functions $\{\boldsymbol{g}_t\}$ with corresponding prediction functions $\{\boldsymbol{f}_t\}$, if $R_{\mathrm{H}}^{\ell}(\boldsymbol{g}_t) \to R_{\mathrm{H}}^{\ell*}$, then $R_{\mathrm{H}}^{0-1}(\boldsymbol{f}_t) \to R_{\mathrm{H}}^*$.*

The proof can be found in Appendix B.5. If the model is flexible enough to include the optimal classifier, according to Theorem 3, we have $R_{\mathrm{H}}^{\ell}(\tilde{\boldsymbol{g}}_{\mathrm{H}}) \to R_{\mathrm{H}}^{\ell*}$. Then, Corollary 1 demonstrates that $R_{\mathrm{H}}^{0-1}(\tilde{\boldsymbol{f}}_{\mathrm{H}}) \to R_{\mathrm{H}}^*$ where $\tilde{\boldsymbol{f}}_{\mathrm{H}}$ is the corresponding prediction function of $\tilde{\boldsymbol{g}}_{\mathrm{H}}$. This indicates that the prediction function obtained by minimizing the corrected risk estimator in Eq. (8) achieves the Bayes risk.

### 3.3 SECOND-ORDER STRATEGY

The first-order strategy is straightforward but does not consider label correlations, which may be incompatible with multi-label data that exhibit semantic dependencies. Therefore, we explore the ranking loss to model the relationship between pairs of labels. The following theorem applies.

**Theorem 4.** *When the binary loss function $\ell$ is symmetric, i.e., $\ell(z, \cdot) + \ell(-z, \cdot) = M$ where $M$ is a non-negative constant, then under the assumption in Lemma 1, the $\ell$-risk w.r.t. the ranking loss in Eq. (4) can be equivalently expressed as*

$$R_{\mathrm{R}}^{\ell}(\boldsymbol{g}) = \sum_{1 \leqslant j < k \leqslant q} \Big( (1 - \pi_j)\mathbb{E}_{p(\boldsymbol{x}|s_j=0)}\left[\ell\left(g_j(\boldsymbol{x}) - g_k(\boldsymbol{x}), 0\right)\right]$$

$$+ (1 - \pi_k)\mathbb{E}_{p(\boldsymbol{x}|s_k=0)}\left[\ell\left(g_j(\boldsymbol{x}) - g_k(\boldsymbol{x}), 1\right)\right] - Mp(y_j = 0, y_k = 0) \Big). \quad (12)$$

The proof can be found in Appendix B.6. Here, the symmetric-loss assumption is often used to ensure statistical consistency of the ranking loss for MLC (Gao & Zhou, 2013). According to Theorem 4, the $\ell$-risk w.r.t. the ranking loss can be expressed as the expectation w.r.t. the conditional density where the $j$-th class label is not regarded as a candidate label. Notably, $Mp(y_j = 0, y_k = 0)$ in Eq. (12) is a constant that does not affect training the classifier, so it can be neglected. Similar to the first-order strategy, an unbiased risk estimator can be obtained using $\mathcal{D}_j$:

$$\hat{R}_{\mathrm{R}}^{\ell}(\boldsymbol{g}) = \sum_{1 \leqslant j < k \leqslant q} \bigg( \frac{1 - \pi_j}{n_j} \sum_{i=1}^{n_j} \ell\left(g_j(\boldsymbol{x}_i^j) - g_k(\boldsymbol{x}_i^j), 0\right)$$

$$+ \frac{1 - \pi_k}{n_k} \sum_{i=1}^{n_k} \ell\left(g_j(\boldsymbol{x}_i^k) - g_k(\boldsymbol{x}_i^k), 1\right) \bigg). \quad (13)$$

To improve generalization performance, we use the flooding regularization technique (Ishida et al., 2020; Liu et al., 2022a; Bae et al., 2024) to mitigate overfitting issues:

$$\tilde{R}_{\mathrm{R}}^{\ell}(\boldsymbol{g}) = \left| \hat{R}_{\mathrm{R}}^{\ell}(\boldsymbol{g}) - \beta \right| + \beta, \tag{14}$$

where $\beta \geqslant 0$ is a hyper-parameter that controls the minimum of the loss value. Then, we can perform ERM by using Eq. (14). The algorithmic details are summarized in Algorithm 1. We also establish consistency and estimation error bounds for the proposed risk estimator in Eq. (14). The following theorem then holds.

**Theorem 5.** *We assume that there exists a positive constant $\gamma$ such that $R_{\mathrm{R}}^{\ell}(\boldsymbol{g}) \geqslant \gamma$. We also assume that $\beta$ is chosen such that $\beta \leqslant \sum_{1 \leqslant j < k \leqslant q} Mp(y_j = 0, y_k = 0)z$. By the assumptions in Theorem 2, the bias of the expectation of the corrected risk estimator w.r.t. the ranking loss has the following lower and upper bounds:*

$$0 \leqslant \mathbb{E}\left[\tilde{R}_{\mathrm{R}}^{\ell}(\boldsymbol{g})\right] - \sum_{j<k} Mp(y_j = 0, y_k = 0) - R_{\mathrm{R}}^{\ell}(\boldsymbol{g}) \leqslant \left(2\beta + 2C_{\ell}(q-1)\sum_{j=1}^{q}(1-\pi_j)\right)\Delta', \tag{15}$$

*where $\Delta' = \exp\left(-2\gamma^2 / \sum_{j=1}^{q}(1-\pi_j)^2(q-1)^2 C_{\ell}^2/n_j\right)$. Furthermore, for any $\delta > 0$, the following inequality holds with probability at least $1 - \delta$:*

$$\left| \tilde{R}_{\mathrm{R}}^{\ell}(\boldsymbol{g}) - \sum_{j<k} Mp(y_j = 0, y_k = 0) - R_{\mathrm{R}}^{\ell}(\boldsymbol{g}) \right| \leqslant \sum_{j=1}^{q}(1-\pi_j)(q-1)C_{\ell}\sqrt{\frac{\ln(2/\delta)}{2n_j}}$$

$$+ \left(2\beta + 2C_{\ell}(q-1)\sum_{j=1}^{q}(1-\pi_j)\right)\Delta'. \tag{16}$$

The proof can be found in Appendix B.7. According to Theorem 5, as $n \to \infty$, the bias between the corrected risk estimator in Eq. (14) and the $\ell$-risk of ranking loss will become a constant. This implies that the minimizer of the corrected risk estimator is equivalent to the desired classifier that minimizes the $\ell$-risk w.r.t. the Hamming loss.

Let $\tilde{\boldsymbol{g}}_{\mathrm{R}} = \arg\min_{\{g_j\} \subseteq \mathcal{G}} \tilde{R}_{\mathrm{R}}^{\ell}(\boldsymbol{g})$ and $\boldsymbol{g}_{\mathrm{R}}^{*} = \arg\min_{\{g_j\} \subseteq \mathcal{G}} R_{\mathrm{R}}^{\ell}(\boldsymbol{g})$ denote the minimizers of the corrected risk estimator and the $\ell$-risk w.r.t. the ranking loss, respectively.

**Theorem 6.** *By the assumptions in Theorem 3 and 5, for any $\delta > 0$, the following inequality holds with probability at least $1 - \delta$:*

$$R_{\mathrm{R}}^{\ell}(\tilde{\boldsymbol{g}}_{\mathrm{R}}) - R_{\mathrm{R}}^{\ell}(\boldsymbol{g}_{\mathrm{R}}^{*}) \leqslant \left(2\beta + 2C_{\ell}(q-1)\sum_{j=1}^{q}(1-\pi_j)\right)\Delta'$$

$$+ \sum_{j=1}^{q}(1-\pi_j)(q-1)C_{\ell}\sqrt{\frac{\ln(1/\delta)}{n_j}} + \sum_{j=1}^{q} 4L_{\ell}(q-1)(1-\pi_j)\mathfrak{R}_{n_j,p_j}(\mathcal{G}). \tag{17}$$

The proof can be found in Appendix B.8. Theorem 6 shows that as $n \to \infty$, $R_{\mathrm{R}}^{\ell}(\tilde{\boldsymbol{g}}_{\mathrm{R}}) \to R_{\mathrm{R}}^{\ell}(\boldsymbol{g}_{\mathrm{R}}^{*})$, since $\Delta' \to 0$ and $\mathfrak{R}_{n_j,p_j}(\mathcal{G}) \to 0$ for all parametric models with a bounded norm (Mohri et al., 2012). This means that the minimizers of Eq. (14) will approach the desired classifiers of the $\ell$-risk w.r.t. the ranking loss when the number of training data increases. Let $R_{\mathrm{R}}^{\ell*} = \inf_{\boldsymbol{g}} R_{\mathrm{R}}^{\ell}(\boldsymbol{g})$ and $R_{\mathrm{R}}^{*} = \inf_{\boldsymbol{f}} R_{\mathrm{R}}^{0-1}(\boldsymbol{f})$ denote the minima of the $\ell$-risk and the risk w.r.t. the ranking loss, respectively. Then we have the following corollary.

**Corollary 2.** *If $\ell$ is a differentiable, symmetric, and non-increasing function such that $\forall y, \ell'(0, y) < 0$ and $\ell(z, y) + \ell(-z, y) = M$, then the $\ell$-risk w.r.t. the ranking loss in Eq. (12) is consistent with the risk w.r.t. the ranking loss in Eq. (3). This means that for any sequences of decision functions $\{\boldsymbol{g}_t\}$ with corresponding prediction functions $\{\boldsymbol{f}_t\}$, if $R_{\mathrm{R}}^{\ell}(\boldsymbol{g}_t) \to R_{\mathrm{R}}^{\ell*}$, then $R_{\mathrm{R}}^{0-1}(\boldsymbol{f}_t) \to R_{\mathrm{R}}^{*}$.*

The proof can be found in Appendix B.9. If the model is very flexible, we have $R_{\mathrm{R}}^{\ell}(\tilde{\boldsymbol{g}}_{\mathrm{R}}) \to R_{\mathrm{R}}^{\ell*}$ according to Theorem 6. Then, Corollary 2 demonstrates that $R_{\mathrm{R}}^{0-1}(\tilde{\boldsymbol{f}}_{\mathrm{R}}) \to R_{\mathrm{R}}^{*}$ where $\tilde{\boldsymbol{f}}_{\mathrm{R}}$ is the corresponding prediction function of $\tilde{\boldsymbol{g}}_{\mathrm{R}}$. This indicates that the prediction function obtained by minimizing Eq. (14) achieves the Bayes risk.

## 4 EXPERIMENTS

In this section, we validate the effectiveness of the proposed approaches with experimental results.

Table 2: Experimental results on real-world benchmark datasets. Lower is better for the *ranking loss*, *one error*, *Hamming loss*, *coverage*; higher is better for the *average precision*.

| | Ranking Loss ↓ | | | | | |
|---|---|---|---|---|---|---|
| Approach | mirflickr | music_emotion | music_style | yeastBP | yeastCC | yeastMF |
| BCE | 0.106 ± 0.008● | 0.244 ± 0.007● | 0.137 ± 0.009 | 0.328 ± 0.013● | 0.206 ± 0.011● | 0.251 ± 0.010● |
| CCMN | 0.106 ± 0.011● | 0.224 ± 0.007● | 0.155 ± 0.012● | 0.328 ± 0.011● | 0.210 ± 0.013● | 0.245 ± 0.011● |
| GDF | 0.159 ± 0.007● | 0.278 ± 0.010● | 0.160 ± 0.008● | 0.501 ± 0.009● | 0.504 ± 0.016● | 0.495 ± 0.029● |
| CTL | 0.130 ± 0.006● | 0.266 ± 0.010● | 0.179 ± 0.008● | 0.498 ± 0.007● | 0.467 ± 0.014● | 0.471 ± 0.026● |
| MLCL | 0.498 ± 0.035● | 0.470 ± 0.046● | **0.130 ± 0.010** | 0.453 ± 0.033● | 0.222 ± 0.047● | 0.231 ± 0.077● |
| COMES-HL | **0.095 ± 0.009** | 0.214 ± 0.005 | 0.132 ± 0.010 | **0.154 ± 0.010** | 0.124 ± 0.011 | 0.173 ± 0.021● |
| COMES-RL | 0.106 ± 0.006● | **0.213 ± 0.003** | 0.147 ± 0.013● | 0.166 ± 0.010● | **0.117 ± 0.009** | **0.151 ± 0.014** |

| | One Error ↓ | | | | | |
|---|---|---|---|---|---|---|
| Approach | mirflickr | music_emotion | music_style | yeastBP | yeastCC | yeastMF |
| BCE | 0.275 ± 0.021● | 0.462 ± 0.015● | 0.345 ± 0.019 | 0.871 ± 0.008● | 0.814 ± 0.019● | 0.886 ± 0.020● |
| CCMN | 0.282 ± 0.030● | 0.385 ± 0.018 | 0.346 ± 0.017 | 0.878 ± 0.016● | 0.823 ± 0.016● | 0.882 ± 0.012● |
| GDF | 0.409 ± 0.027● | 0.531 ± 0.012● | 0.367 ± 0.018● | 0.976 ± 0.006● | 0.971 ± 0.008● | 0.972 ± 0.007● |
| CTL | 0.366 ± 0.017● | 0.469 ± 0.019● | 0.394 ± 0.022● | 0.970 ± 0.006● | 0.964 ± 0.004● | 0.963 ± 0.010● |
| MLCL | 0.810 ± 0.066● | 0.793 ± 0.041● | 0.405 ± 0.068● | 0.961 ± 0.038● | 0.862 ± 0.066● | 0.887 ± 0.066● |
| COMES-HL | **0.171 ± 0.019** | **0.382 ± 0.015** | **0.333 ± 0.012** | **0.641 ± 0.030** | **0.744 ± 0.020** | **0.800 ± 0.023** |
| COMES-RL | 0.206 ± 0.036● | 0.409 ± 0.015● | 0.351 ± 0.021● | 0.808 ± 0.016● | 0.754 ± 0.022 | 0.805 ± 0.020 |

| | Hamming Loss ↓ | | | | | |
|---|---|---|---|---|---|---|
| Approach | mirflickr | music_emotion | music_style | yeastBP | yeastCC | yeastMF |
| BCE | 0.220 ± 0.007● | 0.307 ± 0.007● | 0.186 ± 0.005● | 0.148 ± 0.007● | 0.162 ± 0.007● | 0.153 ± 0.006● |
| CCMN | 0.220 ± 0.006● | 0.284 ± 0.013● | 0.239 ± 0.020● | 0.151 ± 0.007● | 0.163 ± 0.008● | 0.150 ± 0.005● |
| GDF | 0.277 ± 0.007● | 0.374 ± 0.009● | 0.251 ± 0.008● | 0.499 ± 0.016● | 0.489 ± 0.026● | 0.497 ± 0.030● |
| CTL | 0.237 ± 0.006● | 0.349 ± 0.006● | 0.298 ± 0.008● | 0.493 ± 0.009● | 0.499 ± 0.007● | 0.496 ± 0.006● |
| MLCL | 0.601 ± 0.020● | 0.480 ± 0.025● | 0.246 ± 0.019● | 0.881 ± 0.096● | 0.845 ± 0.051● | 0.837 ± 0.024● |
| COMES-HL | **0.164 ± 0.003** | **0.247 ± 0.005** | **0.120 ± 0.006** | 0.073 ± 0.008● | 0.119 ± 0.015● | 0.101 ± 0.005● |
| COMES-RL | 0.186 ± 0.008● | 0.278 ± 0.005 ● | 0.210 ± 0.008● | **0.051 ± 0.001** | **0.045 ± 0.004** | **0.048 ± 0.003** |

| | Coverage ↓ | | | | | |
|---|---|---|---|---|---|---|
| Approach | mirflickr | music_emotion | music_style | yeastBP | yeastCC | yeastMF |
| BCE | 0.212 ± 0.009 | 0.408 ± 0.010● | 0.197 ± 0.011 | 0.437 ± 0.021● | 0.123 ± 0.010● | 0.125 ± 0.011● |
| CCMN | 0.212 ± 0.012 | 0.392 ± 0.010● | 0.216 ± 0.015● | 0.436 ± 0.022● | 0.125 ± 0.012● | 0.123 ± 0.012● |
| GDF | 0.254 ± 0.006● | 0.440 ± 0.010● | 0.220 ± 0.010● | 0.569 ± 0.019● | 0.273 ± 0.018● | 0.242 ± 0.022● |
| CTL | 0.229 ± 0.008● | 0.441 ± 0.014● | 0.240 ± 0.011● | 0.567 ± 0.016● | 0.259 ± 0.017● | 0.231 ± 0.015● |
| MLCL | 0.492 ± 0.036● | 0.596 ± 0.047● | **0.177 ± 0.013** | 0.530 ± 0.072● | 0.137 ± 0.045● | 0.099 ± 0.032● |
| COMES-HL | **0.211 ± 0.008** | 0.379 ± 0.008 | 0.192 ± 0.012 | 0.229 ± 0.016 | 0.070 ± 0.008 | 0.085 ± 0.006● |
| COMES-RL | 0.224 ± 0.008● | **0.377 ± 0.006** | 0.208 ± 0.015● | **0.219 ± 0.015** | **0.070 ± 0.006** | **0.073 ± 0.005** |

| | Average Precision ↑ | | | | | |
|---|---|---|---|---|---|---|
| Approach | mirflickr | music_emotion | music_style | yeastBP | yeastCC | yeastMF |
| BCE | 0.813 ± 0.011● | 0.616 ± 0.009● | 0.738 ± 0.013● | 0.150 ± 0.013● | 0.487 ± 0.016● | 0.379 ± 0.019● |
| CCMN | 0.811 ± 0.016● | 0.660 ± 0.010 | 0.728 ± 0.013● | 0.150 ± 0.012● | 0.479 ± 0.016● | 0.386 ± 0.021● |
| GDF | 0.742 ± 0.013● | 0.574 ± 0.008● | 0.711 ± 0.011● | 0.057 ± 0.002● | 0.135 ± 0.010● | 0.144 ± 0.016● |
| CTL | 0.772 ± 0.009● | 0.600 ± 0.011● | 0.692 ± 0.012● | 0.060 ± 0.002● | 0.154 ± 0.004● | 0.165 ± 0.013● |
| MLCL | 0.446 ± 0.038● | 0.381 ± 0.029● | 0.719 ± 0.035● | 0.082 ± 0.015● | 0.402 ± 0.080● | 0.375 ± 0.124● |
| COMES-HL | **0.843 ± 0.013** | 0.665 ± 0.009 | **0.749 ± 0.010** | **0.458 ± 0.020** | **0.657 ± 0.020** | **0.552 ± 0.023** |
| COMES-RL | 0.818 ± 0.011● | **0.665 ± 0.006** | 0.732 ± 0.013● | 0.315 ± 0.015● | 0.651 ± 0.023 | 0.549 ± 0.019 |

## 4.1 EXPERIMENTAL SETUP

We conducted experiments on both real-world and synthetic PML benchmark datasets. For real-world datasets, we used mirflickr (Huiskes & Lew, 2008), music_emotion (Zhang & Fang, 2020), music_style, yeastBP (Yu et al., 2018a), yeastCC and yeastMF; for synthetic datasets, we used VOC2007 (Everingham et al., 2007), VOC2012 (Everingham et al., 2012), CUB (Wah et al., 2011) and COCO2014 (Lin et al., 2014), where candidate labels were generated by two different data generation processes. Full experimental details are given in Appendix C. Following standard practice (Liu et al., 2023), we evaluated with *ranking loss, one error, Hamming loss, coverage* and *average precision* on real-world sets, and with *mean average precision (mAP)* on synthetic datasets.

For real-world datasets, we used an MLP encoder for all baselines, trained for 200 epochs with a learning rate of 5e-3, weight decay of 1e-4, and the SGD optimizer with cosine decay. For synthetic

Table 3: Classification performance in terms of mAP on synthetic benchmark datasets.

| | VOC2007 | | VOC2012 | | CUB | | COCO2014 | |
|---|---|---|---|---|---|---|---|---|
| Approach | Case-a | Case-b | Case-a | Case-b | Case-a | Case-b | Case-a | Case-b |
| BCE | $40.26 \pm 2.79\bullet$ | $38.87 \pm 1.12\bullet$ | $37.59 \pm 1.29\bullet$ | $41.17 \pm 2.98\bullet$ | $16.30 \pm 0.48\bullet$ | $16.09 \pm 0.14\bullet$ | $26.73 \pm 1.12\bullet$ | $27.10 \pm 0.40\bullet$ |
| CCMN | $40.02 \pm 4.98\bullet$ | $39.84 \pm 1.22\bullet$ | $39.16 \pm 2.34\bullet$ | $42.05 \pm 4.84\bullet$ | $16.51 \pm 0.25\bullet$ | $16.97 \pm 0.90\bullet$ | $25.24 \pm 1.81\bullet$ | $26.79 \pm 1.23\bullet$ |
| GDF | $21.27 \pm 1.03\bullet$ | $20.19 \pm 0.43\bullet$ | $23.58 \pm 2.55\bullet$ | $22.96 \pm 2.87\bullet$ | $12.83 \pm 0.15\bullet$ | $12.77 \pm 0.10\bullet$ | $17.32 \pm 0.62\bullet$ | $15.86 \pm 0.35\bullet$ |
| CTL | $17.05 \pm 0.90\bullet$ | $18.87 \pm 1.86\bullet$ | $19.38 \pm 0.81\bullet$ | $18.51 \pm 0.71\bullet$ | $11.94 \pm 0.23\bullet$ | $11.94 \pm 0.23\bullet$ | $6.31 \pm 0.30\bullet$ | $6.34 \pm 0.14\bullet$ |
| MLCL | $23.42 \pm 1.66\bullet$ | $17.78 \pm 1.18\bullet$ | $15.02 \pm 4.68\bullet$ | $15.00 \pm 3.54\bullet$ | $16.80 \pm 0.04\bullet$ | $17.92 \pm 0.10\bullet$ | $10.59 \pm 0.63\bullet$ | $10.67 \pm 0.81\bullet$ |
| COMES-HL | $42.33 \pm 1.74\bullet$ | $42.43 \pm 4.17\bullet$ | $48.72 \pm 1.08\bullet$ | $47.93 \pm 1.05\bullet$ | $\mathbf{18.94 \pm 0.30}$ | $\mathbf{18.95 \pm 0.39}$ | $\mathbf{33.62 \pm 0.57}$ | $\mathbf{32.76 \pm 1.45}$ |
| COMES-RL | $\mathbf{51.46 \pm 3.09}$ | $\mathbf{49.42 \pm 4.27}$ | $\mathbf{53.26 \pm 0.74}$ | $\mathbf{52.29 \pm 4.15}$ | $17.50 \pm 0.33\bullet$ | $17.34 \pm 0.03\bullet$ | $27.98 \pm 0.30\bullet$ | $28.69 \pm 1.62\bullet$ |

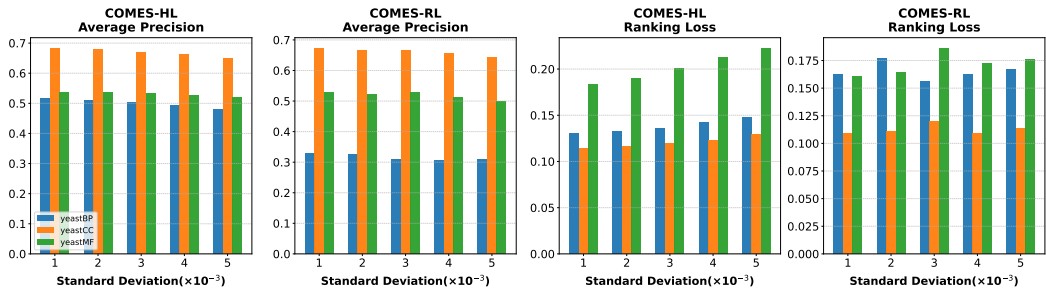

Figure 2: Classification performance with inaccurate class priors on different datasets.

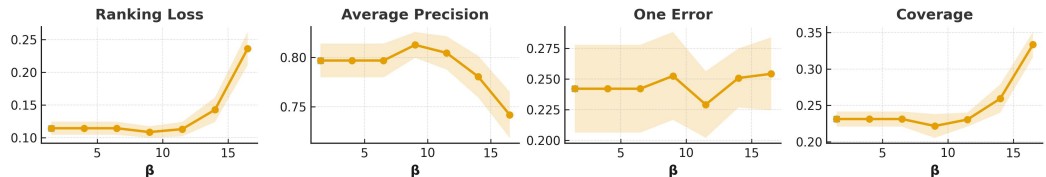

Figure 3: Sensitivity analysis w.r.t. $\beta$ on the mirflickr dataset.

image datasets, we adopted a ResNet-50 backbone pretrained on ImageNet (Deng et al., 2009), trained for 30 epochs with a learning rate of 1e-4 using the Adam optimizer. For fair comparisons, we used the same setup across all baselines. We assumed that the class priors were accessible to the learning algorithm. We instantiated $\ell$ with the binary cross-entropy loss for COMES-HL and the sigmoid loss for COMES-RL. We performed ten-fold cross-validation on real-world datasets. This means we used nine folds for training and one fold for testing. Then, we recorded the mean accuracy and standard deviation. For the synthetic datasets, we generated synthetic labels three times and recorded the mean accuracy and standard deviation. Finally, we conducted paired t-tests at a 0.05 significance level.

## 4.2 EXPERIMENTAL RESULTS

Tables 2 and 3 summarize results on real-world and synthetic datasets, respectively. Here ● indicates that the best method is significantly better than its competitor (paired $t$-test at 0.05 significance level). We observe that both instantiations of COMES consistently outperform other baselines across various datasets, clearly validating the effectiveness of our proposed approaches. We attribute this to: (1) our data generation assumptions are more realistic and better match statistics of real-world datasets; (2) our risk-correction approaches effectively mitigates overfitting, an issue overlooked by previous unbiased methods (Xie & Huang, 2023).

## 4.3 SENSITIVITY ANALYSIS

**Influence of Inaccurate Class Priors.** To investigate the influence of inaccurate class priors, we added Gaussian noise $\epsilon \sim \mathcal{N}(0, \sigma^2)$ to each class prior $\pi_j$. The experimental results are shown in Figure 2. We observe that COMES-HL is more sensitive to inaccurate class priors. Overall performance remains stable within a reasonable range of class priors, but may degrade when the priors become highly inaccurate.

**Influence of $\beta$.** We also investigated the influence of the hyperparameter $\beta$ for the flooding regularization used in COMES-RL. From Figure 3, we observe that the performance of COMES-RL is rather stable when $\beta$ is set within a reasonable range on the mirflickr dataset. During our experiments, we found that the performance was already competitive by setting $\beta = 0$ on many datasets. However, the performance may degrade when $\beta$ is set to a large value. This also matches our theoretical results in Theorem 5, where consistency holds when $\beta$ is not too large.

## 5  CONCLUSION

In this paper, we rethought MLC under inexact supervision by proposing a novel framework. We proposed two instantiations of risk estimators w.r.t. the Hamming loss and ranking loss, two widely used evaluation metrics for MLC, respectively. We also introduced risk-correction approaches to improve generalization performance with theoretical guarantees. Extensive experiments on ten real-world and synthetic benchmark datasets validated the effectiveness of the proposed approaches. A limitation of this work is that we consider the generation process to be independent of the instances. In the future, it is promising to extend our proposed methodologies to instance-dependent settings.

ACKNOWLEDGMENTS

WW was supported by the Junior Research Associate (JRA) program of RIKEN. MS was supported by JST ASPIRE Grant Number JPMJAP2405.

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

THE USE OF LARGE LANGUAGE MODELS (LLMS)

We used LLMs to check the manuscript for typos and grammatical errors.

# A   MORE DETAILS ABOUT THE ALGORITHM

## A.1   ALGORITHMIC PSEUDO-CODE

---

**Algorithm 1** COMES-HL and COMES-RL

---

**Input:** Multi-label classifiers $g$, PML dataset $\mathcal{D}$, epoch number $T_{\max}$, iteration number $I_{\max}$.

1: **for** $t = 1, 2, \ldots, T_{\max}$ **do**
2:     **Shuffle** $\mathcal{D}$;
3:     **for** $j = 1, \ldots, I_{\max}$ **do**
4:         **Fetch** mini-batch $\mathcal{D}_j$ from $\mathcal{D}$;
5:         **Forward** $\mathcal{D}$ and get the outputs of $g$;
6:         **if** using the COMES-HL algorithm **then**
7:             **Calculating** the loss based on Eq. (8);
8:         **else if** using the COMES-RL algorithm **then**
9:             **Calculating** the loss based on Eq. (14);
10:        **end if**
11:        **Update** $g$ using a stochastic optimizer to minimize the loss;
12:     **end for**
13: **end for**

**Output:** $g$.

---

## A.2   CLASS PRIOR ESTIMATION

We can use any off-the-shelf mixture proportion estimation algorithm to estimate the class priors (Ramaswamy et al., 2016; Garg et al., 2021; Yao et al., 2022; Zhu et al., 2023), which are mainly designed to estimate the class prior with positive and unlabeled data for binary classification. Specifically, we generate negative and unlabeled datasets according to Eq. (6) and then apply the mixture proportion estimation algorithm. The algorithmic details are summarized in Algorithm 2.

---

**Algorithm 2** Class-prior Estimation

---

**Input:** Mixture proportion estimation algorithm $\mathcal{A}$, PML dataset $\mathcal{D}$.

1: **for** $k \in \mathcal{Y}$ **do**
2:     **Generate** unlabeled and negative datasets according to Eq. (6);
3:     **Estimate** the value of $(1 - \pi_k)$ by using $\mathcal{A}$ and interchanging the positive and negative classes;
4: **end for**

**Output:** Class priors $\pi_k$ ($k \in \mathcal{Y}$).

---

# B   PROOFS

## B.1   PROOF OF LEMMA 1

According to the definition of PML, when $s_j = 0$, the $j$-th class is impossible to be a relevant label, and we have $p(j \notin Y | \boldsymbol{x}, s_j = 0) = p(j \notin Y | s_j = 0) = 1$. Therefore, on one hand, we have

$$p(\boldsymbol{x}|s_j = 0, j \notin Y) = \frac{p(\boldsymbol{x}|s_j = 0)\, p(j \notin Y|\boldsymbol{x}, s_j = 0)}{p(j \notin Y|s_j = 0)} = p(\boldsymbol{x}|s_j = 0).$$

On the other hand, we have

$$p(\boldsymbol{x}|s_j = 0, j \notin Y) = \frac{p(\boldsymbol{x}|j \notin Y)p(s_j = 0|\boldsymbol{x}, j \notin Y)}{p(s_j = 0|j \notin Y)} = p(\boldsymbol{x}|j \notin Y),$$

where the second equation is due to $p(s_j = 0|\boldsymbol{x}, j \notin Y) = p(s_j = 0|j \notin Y) = p_j$. The proof is completed. $\qquad\square$

## B.2 Proof of Theorem 1

$$
\begin{aligned}
R_{\mathrm{H}}^{\ell}(\boldsymbol{g}) &= \mathbb{E}_{p(\boldsymbol{x},Y)} \left[ \frac{1}{q} \sum\nolimits_{j=1}^{q} \ell\left(g_j\left(\boldsymbol{x}\right), y_j\right) \right] \\
&= \int \sum\nolimits_Y \frac{1}{q} \sum\nolimits_{j=1}^{q} \ell\left(g_j\left(\boldsymbol{x}\right), y_j\right) p\left(\boldsymbol{x}, Y\right) \mathrm{d}\boldsymbol{x} \\
&= \int \frac{1}{q} \sum\nolimits_{j=1}^{q} \sum\nolimits_{y_j=0}^{1} \sum\nolimits_{Y'=Y\backslash j} \ell\left(g_j\left(\boldsymbol{x}\right), y_j\right) p(\boldsymbol{x}, y_j) p\left(Y'|\boldsymbol{x}, y_j\right) \mathrm{d}\boldsymbol{x} \\
&= \int \frac{1}{q} \sum\nolimits_{j=1}^{q} \sum\nolimits_{y_j=0}^{1} \ell\left(g_j\left(\boldsymbol{x}\right), y_j\right) p(\boldsymbol{x}, y_j) \sum\nolimits_{Y'=Y\backslash j} p\left(Y'|\boldsymbol{x}, y_j\right) \mathrm{d}\boldsymbol{x} \\
&= \int \frac{1}{q} \sum\nolimits_{j=1}^{q} \sum\nolimits_{y_j=0}^{1} \ell\left(g_j\left(\boldsymbol{x}\right), y_j\right) p(\boldsymbol{x}, y_j) \mathrm{d}\boldsymbol{x} \\
&= \int \frac{1}{q} \sum\nolimits_{j=1}^{q} \ell\left(g_j\left(\boldsymbol{x}\right), 1\right) p(\boldsymbol{x}, y_j=1) \mathrm{d}\boldsymbol{x} + \int \frac{1}{q} \sum\nolimits_{j=1}^{q} \ell\left(g_j\left(\boldsymbol{x}\right), 0\right) p\left(\boldsymbol{x}, y_j=0\right) \mathrm{d}\boldsymbol{x} \\
&= \int \frac{1}{q} \sum\nolimits_{j=1}^{q} \ell\left(g_j\left(\boldsymbol{x}\right), 1\right) p(\boldsymbol{x}) \mathrm{d}\boldsymbol{x} - \int \frac{1}{q} \sum\nolimits_{j=1}^{q} \ell\left(g_j\left(\boldsymbol{x}\right), 1\right) p\left(\boldsymbol{x}, y_j=0\right) \mathrm{d}\boldsymbol{x} \\
&\quad + \int \frac{1}{q} \sum\nolimits_{j=1}^{q} \ell\left(g_j\left(\boldsymbol{x}\right), 0\right) p\left(\boldsymbol{x}, y_j=0\right) \mathrm{d}\boldsymbol{x} \\
&= \int \frac{1}{q} \sum\nolimits_{j=1}^{q} \ell\left(g_j\left(\boldsymbol{x}\right), 1\right) p(\boldsymbol{x}) \mathrm{d}\boldsymbol{x} + \int \frac{1}{q} \sum\nolimits_{j=1}^{q} \left(\ell\left(g_j\left(\boldsymbol{x}\right), 0\right) - \ell\left(g_j\left(\boldsymbol{x}\right), 1\right)\right)\left(1 - \pi_j\right) p\left(\boldsymbol{x}|y_j=0\right) \mathrm{d}\boldsymbol{x} \\
&= \mathbb{E}_{p(\boldsymbol{x})} \left[ \frac{1}{q} \sum\nolimits_{j=1}^{q} \ell\left(g_j\left(\boldsymbol{x}\right), 1\right) \right] + \mathbb{E}_{p(\boldsymbol{x}|y_j=0)} \left[ \frac{1}{q} \sum\nolimits_{j=1}^{q} \left(1 - \pi_j\right)\left(\ell\left(g_j\left(\boldsymbol{x}\right), 0\right) - \ell\left(g_j\left(\boldsymbol{x}\right), 1\right)\right) \right] \\
&= \mathbb{E}_{p(\boldsymbol{x})} \left[ \frac{1}{q} \sum\nolimits_{j=1}^{q} \ell\left(g_j\left(\boldsymbol{x}\right), 1\right) \right] + \mathbb{E}_{p(\boldsymbol{x}|s_j=0)} \left[ \frac{1}{q} \sum\nolimits_{j=1}^{q} \left(1 - \pi_j\right)\left(\ell\left(g_j\left(\boldsymbol{x}\right), 0\right) - \ell\left(g_j\left(\boldsymbol{x}\right), 1\right)\right) \right],
\end{aligned}
$$

where the last equation is by Lemma 1. The proof is completed. $\qquad\square$

## B.3 Proof of Theorem 2

Let

$$
\mathfrak{D}_j^{+}(g_j) = \left\{ (\mathcal{D}_{\mathrm{U}}, \mathcal{D}_j) | \frac{1}{n} \sum\nolimits_{i=1}^{n} \ell\left(g_j\left(\boldsymbol{x}_i^{\mathrm{U}}\right), 1\right) - \frac{1 - \pi_j}{n_j} \sum\nolimits_{i=1}^{n_j} \ell\left(g_j\left(\boldsymbol{x}_i^{j}\right), 1\right) > 0 \right\}
$$

and

$$
\mathfrak{D}_j^{-}(g_j) = \left\{ (\mathcal{D}_{\mathrm{U}}, \mathcal{D}_j) | \frac{1}{n} \sum\nolimits_{i=1}^{n} \ell\left(g_j\left(\boldsymbol{x}_i^{\mathrm{U}}\right), 1\right) - \frac{1 - \pi_j}{n_j} \sum\nolimits_{i=1}^{n_j} \ell\left(g_j\left(\boldsymbol{x}_i^{j}\right), 1\right) \leqslant 0 \right\}
$$

denote the set of data pairs with positive and negative empirical losses, respectively. Then, we have the following lemma.

**Lemma 2.** *The probability measure of $\mathfrak{D}_j^{-}(g_j)$ can be bounded as follows:*

$$
\mathbb{P}\left(\mathfrak{D}_j^{-}(g_j)\right) \leqslant \exp\left( \frac{-2\alpha^2}{C_{\ell}^2/n + (1 - \pi_j)^2 C_{\ell}^2/n_j} \right). \tag{18}
$$

*Proof.* Let

$$
p\left(\mathcal{D}_{\mathrm{U}}\right) = p(\boldsymbol{x}_1^{\mathrm{U}}) p(\boldsymbol{x}_2^{\mathrm{U}}) \cdots p(\boldsymbol{x}_n^{\mathrm{U}}) \quad \text{and} \quad p\left(\mathcal{D}_j\right) = p(\boldsymbol{x}_1^{j}) p(\boldsymbol{x}_2^{j}) \cdots p(\boldsymbol{x}_{n_j}^{j})
$$

denote the densities of $\mathcal{D}_{\mathrm{U}}$ and $\mathcal{D}_j$, respectively. Then, the joint density of $(\mathcal{D}_{\mathrm{U}}, \mathcal{D}_j)$ is

$$
p\left(\mathcal{D}_{\mathrm{U}}, \mathcal{D}_j\right) = p\left(\mathcal{D}_{\mathrm{U}}\right) p\left(\mathcal{D}_j\right).
$$

Then, the probability measure of $\mathfrak{D}_j^-(g_j)$ can be expressed as

$$
\begin{aligned}
\mathbb{P}\left(\mathfrak{D}_j^-(g_j)\right) &= \int_{(\mathcal{D}_\mathrm{U},\mathcal{D}_j)\in\mathfrak{D}_j^-(g_j)} p\left(\mathcal{D}_\mathrm{U},\mathcal{D}_j\right) \mathrm{d}\left(\mathcal{D}_\mathrm{U},\mathcal{D}_j\right) \\
&= \int_{(\mathcal{D}_\mathrm{U},\mathcal{D}_j)\in\mathfrak{D}_j^-(g_j)} p\left(\mathcal{D}_\mathrm{U},\mathcal{D}_j\right) \mathrm{d}\boldsymbol{x}_1^\mathrm{U}\,\mathrm{d}\boldsymbol{x}_2^\mathrm{U}\cdots\mathrm{d}\boldsymbol{x}_n^\mathrm{U}\,\mathrm{d}\boldsymbol{x}_1^j\,\mathrm{d}\boldsymbol{x}_2^j\cdots\mathrm{d}\boldsymbol{x}_{n_j}^j
\end{aligned}
$$

When an instance in $\mathcal{D}_\mathrm{U}$ is replaced by another instance, the value of $\sum_{i=1}^n \ell\left(g_j\left(\boldsymbol{x}_i^\mathrm{U}\right),1\right)/n - (1-\pi_j)\sum_{i=1}^{n_j}\ell\left(g_j\left(\boldsymbol{x}_i^j\right),1\right)/n_j$ changes no more than $C_\ell/n$. When an instance in $\mathcal{D}_j$ is replaced by another instance, the value of $\sum_{i=1}^n \ell\left(g_j\left(\boldsymbol{x}_i^\mathrm{U}\right),1\right)/n - (1-\pi_j)\sum_{i=1}^{n_j}\ell\left(g_j\left(\boldsymbol{x}_i^j\right),1\right)/n_j$ changes no more than $(1-\pi_j)C_\ell/n_j$. Therefore, by applying the McDiarmid's inequality, we can obtain the following inequality:

$$
\begin{aligned}
&p\left(\pi_j\mathbb{E}_{p(\boldsymbol{x}|y_j=1)}\left[\ell\left(g_j\left(\boldsymbol{x}\right),1\right)\right] - \left(\frac{1}{n}\sum_{i=1}^n \ell\left(g_j\left(\boldsymbol{x}_i^\mathrm{U}\right),1\right) - \frac{1-\pi_j}{n_j}\sum_{i=1}^{n_j}\ell\left(g_j\left(\boldsymbol{x}_i^j\right),1\right)\right) \geqslant \alpha\right) \\
&\leqslant \exp\left(\frac{-2\alpha^2}{C_\ell^2/n + (1-\pi_j)^2 C_\ell^2/n_j}\right).
\end{aligned}
$$

Then we have

$$
\begin{aligned}
\mathbb{P}\left(\mathfrak{D}_j^-(g_j)\right) &= p\left(\frac{1}{n}\sum_{i=1}^n \ell\left(g_j\left(\boldsymbol{x}_i^\mathrm{U}\right),1\right) - \frac{1-\pi_j}{n_j}\sum_{i=1}^{n_j}\ell\left(g_j\left(\boldsymbol{x}_i^j\right),1\right) \leqslant 0\right) \\
&\leqslant p\left(\frac{1}{n}\sum_{i=1}^n \ell\left(g_j\left(\boldsymbol{x}_i^\mathrm{U}\right),1\right) - \frac{1-\pi_j}{n_j}\sum_{i=1}^{n_j}\ell\left(g_j\left(\boldsymbol{x}_i^j\right),1\right) \leqslant \pi_j\mathbb{E}_{p(\boldsymbol{x}|y_j=1)}\left[\ell\left(g_j\left(\boldsymbol{x}\right),1\right)\right] - \alpha\right) \\
&\leqslant \exp\left(\frac{-2\alpha^2}{C_\ell^2/n + (1-\pi_j)^2 C_\ell^2/n_j}\right),
\end{aligned}
$$

which concludes the proof. $\qquad\square$

Then, we provide the proof of Theorem 2.

*Proof of Theorem 2.* To begin with, we have

$$
\mathbb{E}\left[\tilde{R}_\mathrm{H}^\ell(\boldsymbol{g})\right] - R_\mathrm{H}^\ell(\boldsymbol{g}) = \mathbb{E}\left[\tilde{R}_\mathrm{H}^\ell(\boldsymbol{g}) - \hat{R}_\mathrm{H}^\ell(\boldsymbol{g})\right] \geqslant 0.
$$

Besides, we have

$$
\begin{aligned}
&\left|\frac{1}{n}\sum_{i=1}^n \ell\left(g_j\left(\boldsymbol{x}_i^\mathrm{U}\right),1\right) - \frac{1-\pi_j}{n_j}\sum_{i=1}^{n_j}\ell\left(g_j\left(\boldsymbol{x}_i^j\right),1\right)\right| \\
&\leqslant \left|\frac{1}{n}\sum_{i=1}^n \ell\left(g_j\left(\boldsymbol{x}_i^\mathrm{U}\right),1\right)\right| + \left|\frac{1-\pi_j}{n_j}\sum_{i=1}^{n_j}\ell\left(g_j\left(\boldsymbol{x}_i^j\right),1\right)\right| \\
&\leqslant (2-\pi_j)C_\ell.
\end{aligned}
$$

Then,

$$
\mathbb{E}\left[\tilde{R}_{\mathrm{H}}^{\ell}(\boldsymbol{g})\right] - R_{\mathrm{H}}^{\ell}(\boldsymbol{g})
$$

$$
= \mathbb{E}\left[\tilde{R}_{\mathrm{H}}^{\ell}(\boldsymbol{g}) - \hat{R}_{\mathrm{H}}^{\ell}(\boldsymbol{g})\right]
$$

$$
= \frac{1}{q}\sum_{j=1}^{q}\int_{(\mathcal{D}_{\mathrm{U}},\mathcal{D}_{j})\in\mathfrak{D}_{j}^{-}(g_{j})}\left(\left|\frac{1}{n}\sum_{i=1}^{n}\ell\left(g_{j}\left(\boldsymbol{x}_{i}^{\mathrm{U}}\right),1\right) - \frac{1-\pi_{j}}{n_{j}}\sum_{i=1}^{n_{j}}\ell\left(g_{j}\left(\boldsymbol{x}_{i}^{j}\right),1\right)\right|\right.
$$

$$
\left. - \left(\frac{1}{n}\sum_{i=1}^{n}\ell\left(g_{j}\left(\boldsymbol{x}_{i}^{\mathrm{U}}\right),1\right) - \frac{1-\pi_{j}}{n_{j}}\sum_{i=1}^{n_{j}}\ell\left(g_{j}\left(\boldsymbol{x}_{i}^{j}\right),1\right)\right)\right)p\left(\mathcal{D}_{\mathrm{U}},\mathcal{D}_{j}\right)\mathrm{d}\left(\mathcal{D}_{\mathrm{U}},\mathcal{D}_{j}\right)
$$

$$
\leqslant \frac{1}{q}\sum_{j=1}^{q}\sup_{(\mathcal{D}_{\mathrm{U}},\mathcal{D}_{j})\in\mathfrak{D}_{j}^{-}(g_{j})}\left(\left|\frac{1}{n}\sum_{i=1}^{n}\ell\left(g_{j}\left(\boldsymbol{x}_{i}^{\mathrm{U}}\right),1\right) - \frac{1-\pi_{j}}{n_{j}}\sum_{i=1}^{n_{j}}\ell\left(g_{j}\left(\boldsymbol{x}_{i}^{j}\right),1\right)\right|\right.
$$

$$
\left. - \left(\frac{1}{n}\sum_{i=1}^{n}\ell\left(g_{j}\left(\boldsymbol{x}_{i}^{\mathrm{U}}\right),1\right) - \frac{1-\pi_{j}}{n_{j}}\sum_{i=1}^{n_{j}}\ell\left(g_{j}\left(\boldsymbol{x}_{i}^{j}\right),1\right)\right)\right)\int_{(\mathcal{D}_{\mathrm{U}},\mathcal{D}_{j})\in\mathfrak{D}_{j}^{-}(g_{j})}p\left(\mathcal{D}_{\mathrm{U}},\mathcal{D}_{j}\right)\mathrm{d}\left(\mathcal{D}_{\mathrm{U}},\mathcal{D}_{j}\right)
$$

$$
\leqslant \frac{1}{q}\sum_{j=1}^{q}(4-2\pi_{j})C_{\ell}\mathbb{P}\left(\mathfrak{D}_{j}^{-}(g_{j})\right)
$$

$$
\leqslant \frac{1}{q}\sum_{j=1}^{q}(4-2\pi_{j})C_{\ell}\exp\left(\frac{-2\alpha^{2}}{C_{\ell}^{2}/n + (1-\pi_{j})^{2}C_{\ell}^{2}/n_{j}}\right)
$$

$$
= \frac{1}{q}\sum_{j=1}^{q}(4-2\pi_{j})C_{\ell}\Delta_{j},
$$

which concludes the proof of the first part of the theorem. Then, we provide an upper bound for $\left|\mathbb{E}\left[\tilde{R}_{\mathrm{H}}^{\ell}(\boldsymbol{g})\right] - \tilde{R}_{\mathrm{H}}^{\ell}(\boldsymbol{g})\right|$. When an instance in $\mathcal{D}_{\mathrm{U}}$ is replaced by another instance, the value of $\tilde{R}_{\mathrm{H}}^{\ell}(\boldsymbol{g})$ changes at most $C_{\ell}/n$; when an instance in $\mathcal{D}_{j}$ is replaced by another instance, the value of $\tilde{R}_{\mathrm{H}}^{\ell}(\boldsymbol{g})$ changes at most $(2-2\pi_{j})C_{\ell}/(qn_{j})$. By applying McDiarmid's inequality, we have the following inequalities with probability at least $1-\delta/2$:

$$
\mathbb{E}\left[\tilde{R}_{\mathrm{H}}^{\ell}(\boldsymbol{g})\right] - \tilde{R}_{\mathrm{H}}^{\ell}(\boldsymbol{g}) \leqslant C_{\ell}\sqrt{\frac{\ln(2/\delta)}{2n}} + \sum_{j=1}^{q}\frac{(2-2\pi_{j})C_{\ell}}{q}\sqrt{\frac{\ln(2/\delta)}{2n_{j}}},
$$

$$
\tilde{R}_{\mathrm{H}}^{\ell}(\boldsymbol{g}) - \mathbb{E}\left[\tilde{R}_{\mathrm{H}}^{\ell}(\boldsymbol{g})\right] \leqslant C_{\ell}\sqrt{\frac{\ln(2/\delta)}{2n}} + \sum_{j=1}^{q}\frac{(2-2\pi_{j})C_{\ell}}{q}\sqrt{\frac{\ln(2/\delta)}{2n_{j}}},
$$

where we use the inequality that $\sqrt{a+b}\leqslant\sqrt{a}+\sqrt{b}$. Therefore, the following inequality holds with probability at least $1-\delta$:

$$
\left|\mathbb{E}\left[\tilde{R}_{\mathrm{H}}^{\ell}(\boldsymbol{g})\right] - \tilde{R}_{\mathrm{H}}^{\ell}(\boldsymbol{g})\right| \leqslant C_{\ell}\sqrt{\frac{\ln(2/\delta)}{2n}} + \sum_{j=1}^{q}\frac{(2-2\pi_{j})C_{\ell}}{q}\sqrt{\frac{\ln(2/\delta)}{2n_{j}}}.
$$

Finally, we have

$$
\left|\tilde{R}_{\mathrm{H}}^{\ell}(\boldsymbol{g}) - R_{\mathrm{H}}^{\ell}(\boldsymbol{g})\right|
$$

$$
= \left|\tilde{R}_{\mathrm{H}}^{\ell}(\boldsymbol{g}) - \mathbb{E}\left[\tilde{R}_{\mathrm{H}}^{\ell}(\boldsymbol{g})\right] + \mathbb{E}\left[\tilde{R}_{\mathrm{H}}^{\ell}(\boldsymbol{g})\right] - R_{\mathrm{H}}^{\ell}(\boldsymbol{g})\right|
$$

$$
\leqslant \left|\tilde{R}_{\mathrm{H}}^{\ell}(\boldsymbol{g}) - \mathbb{E}\left[\tilde{R}_{\mathrm{H}}^{\ell}(\boldsymbol{g})\right]\right| + \left|\mathbb{E}\left[\tilde{R}_{\mathrm{H}}^{\ell}(\boldsymbol{g})\right] - R_{\mathrm{H}}^{\ell}(\boldsymbol{g})\right|
$$

$$
= \left|\tilde{R}_{\mathrm{H}}^{\ell}(\boldsymbol{g}) - \mathbb{E}\left[\tilde{R}_{\mathrm{H}}^{\ell}(\boldsymbol{g})\right]\right| + \mathbb{E}\left[\tilde{R}_{\mathrm{H}}^{\ell}(\boldsymbol{g})\right] - R_{\mathrm{H}}^{\ell}(\boldsymbol{g})
$$

$$
\leqslant \frac{1}{q}\sum_{j=1}^{q}\left((4-2\pi_{j})C_{\ell}\exp\left(\frac{-2\alpha^{2}}{C_{\ell}^{2}/n + (1-\pi_{j})^{2}C_{\ell}^{2}/n_{j}}\right) + \frac{(2-2\pi_{j})C_{\ell}}{q}\sqrt{\frac{\ln(2/\delta)}{2n_{j}}}\right) + C_{\ell}\sqrt{\frac{\ln(2/\delta)}{2n}}
$$

$$
= \frac{1}{q}\sum_{j=1}^{q}\left((4-2\pi_{j})C_{\ell}\Delta_{j} + \frac{(2-2\pi_{j})C_{\ell}}{q}\sqrt{\frac{\ln(2/\delta)}{2n_{j}}}\right) + C_{\ell}\sqrt{\frac{\ln(2/\delta)}{2n}},
$$

which concludes the proof. □

### B.4 PROOF OF THEOREM 3

**Definition 1** (Rademacher complexity). Let $\mathcal{X}_n = \{\boldsymbol{x}_1, \ldots \boldsymbol{x}_n\}$ denote $n$ i.i.d. random variables drawn from a probability distribution with density $p(\boldsymbol{x})$, $\mathcal{G} = \{g_k : \mathcal{X} \mapsto \mathbb{R}\}$ denote a class of measurable functions of model outputs for the $k$,-th class, and $\boldsymbol{\sigma} = (\sigma_1, \sigma_2, \ldots, \sigma_n)$ denote Rademacher variables taking values from $\{+1, -1\}$ uniformly. Then, the (expected) Rademacher complexity of $\mathcal{G}$ is defined as

$$\mathfrak{R}_{n,p}(\mathcal{G}) = \mathbb{E}_{\mathcal{X}_n} \mathbb{E}_{\boldsymbol{\sigma}} \left[ \sup_{g_j \in \mathcal{G}} \frac{1}{n} \sum\nolimits_{i=1}^{n} \sigma_i g_j(\boldsymbol{x}_i) \right]. \tag{19}$$

We also introduce an alternative definition of Rademacher complexity:

$$\mathfrak{R}'_{n,p}(\mathcal{G}) = \mathbb{E}_{\mathcal{X}_n} \mathbb{E}_{\boldsymbol{\sigma}} \left[ \sup_{g_j \in \mathcal{G}} \left| \frac{1}{n} \sum\nolimits_{i=1}^{n} \sigma_i g_j(\boldsymbol{x}_i) \right| \right]. \tag{20}$$

Then, we introduce the following lemmas.

**Lemma 3.** *Without any composition, for any $\mathcal{G}$, we have $\mathfrak{R}'_{n,p}(\mathcal{G}) \geqslant \mathfrak{R}_{n,p}(\mathcal{G})$. If $\mathcal{G}$ is closed under negation, we have $\mathfrak{R}'_{n,p}(\mathcal{G}) = \mathfrak{R}_{n,p}(\mathcal{G})$.*

**Lemma 4** (Theorem 4.12 in Ledoux & Talagrand, 1991). *If $\ell : \mathbb{R} \times \{0, 1\} \to \mathbb{R}$ is a Lipschitz continuous function with a Lipschitz constant $L_\ell$ and satisfies $\forall y, \ell(0, y) = 0$, we have*

$$\mathfrak{R}'_{n,p}(\ell \circ \mathcal{G}) \leqslant 2 L_\ell \mathfrak{R}'_{n,p}(\mathcal{G}),$$

*where $\ell \circ \mathcal{G} = \{\ell \circ g_j | g_j \in \mathcal{G}\}$.*

Then, we provide the following lemma.

**Lemma 5.** *Based on the above assumptions, for any $\delta > 0$, the following inequality holds with probability at least $1 - \delta$:*

$$\sup_{g_1, g_2, \ldots, g_q \in \mathcal{G}} \left| R_{\mathrm{H}}^\ell(\boldsymbol{g}) - \tilde{R}_{\mathrm{H}}^\ell(\boldsymbol{g}) \right| \leqslant \frac{4 L_\ell}{q} \sum\nolimits_{j=1}^{q} \mathfrak{R}'_{n,p}(\mathcal{G}) + \frac{8(1 - \pi_j) L_\ell}{q} \sum\nolimits_{j=1}^{q} \mathfrak{R}'_{n_j, p_j}(\mathcal{G})$$

$$+ \frac{1}{q} \sum\nolimits_{j=1}^{q} (4 - 2\pi_j) C_\ell \Delta_j + C_\ell \sqrt{\frac{\ln(1/\delta)}{2n}} + \sum\nolimits_{j=1}^{q} \frac{(2 - 2\pi_j) C_\ell}{q} \sqrt{\frac{\ln(1/\delta)}{2n_j}}.$$

*Proof.* When an instance in $\mathcal{D}_{\mathrm{U}}$ is replaced by another instance, the value of $\sup_{g_1, g_2, \ldots, g_q \in \mathcal{G}} \left| \mathbb{E} \left[ \tilde{R}_{\mathrm{H}}^\ell(\boldsymbol{g}) \right] - \tilde{R}_{\mathrm{H}}^\ell(\boldsymbol{g}) \right|$ changes at most $C_\ell / n$; when an instance in $\mathcal{D}_j$ is replaced by another instance, the value of $\sup_{g_1, g_2, \ldots, g_q \in \mathcal{G}} \left| \mathbb{E} \left[ \tilde{R}_{\mathrm{H}}^\ell(\boldsymbol{g}) \right] - \tilde{R}_{\mathrm{H}}^\ell(\boldsymbol{g}) \right|$ changes at most $(2 - 2\pi_j) C_\ell / (q n_j)$. By applying McDiarmid's inequality, we have the following inequality with probability at least $1 - \delta$:

$$\sup_{g_1, g_2, \ldots, g_q \in \mathcal{G}} \left| \mathbb{E} \left[ \tilde{R}_{\mathrm{H}}^\ell(\boldsymbol{g}) \right] - \tilde{R}_{\mathrm{H}}^\ell(\boldsymbol{g}) \right| - \mathbb{E} \left[ \sup_{g_1, g_2, \ldots, g_q \in \mathcal{G}} \left| \mathbb{E} \left[ \tilde{R}_{\mathrm{H}}^\ell(\boldsymbol{g}) \right] - \tilde{R}_{\mathrm{H}}^\ell(\boldsymbol{g}) \right| \right]$$

$$\leqslant C_\ell \sqrt{\frac{\ln(1/\delta)}{2n}} + \sum\nolimits_{j=1}^{q} \frac{(2 - 2\pi_j) C_\ell}{q} \sqrt{\frac{\ln(1/\delta)}{2n_j}}. \tag{21}$$

For ease of notations, let $\overline{\mathcal{D}} = \mathcal{D}_{\mathrm{U}} \bigcup \mathcal{D}_1 \bigcup \mathcal{D}_2 \ldots \bigcup \mathcal{D}_q$ denote set of all the data. We have

$$\mathbb{E} \left[ \sup_{g_1, g_2, \ldots, g_q \in \mathcal{G}} \left| \mathbb{E} \left[ \tilde{R}_{\mathrm{H}}^\ell(\boldsymbol{g}) \right] - \tilde{R}_{\mathrm{H}}^\ell(\boldsymbol{g}) \right| \right]$$

$$= \mathbb{E}_{\overline{\mathcal{D}}} \left[ \sup_{g_1, g_2, \ldots, g_q \in \mathcal{G}} \left| \mathbb{E}_{\overline{\mathcal{D}}'} \left[ \tilde{R}_{\mathrm{H}}^\ell(\boldsymbol{g}) \right] - \tilde{R}_{\mathrm{H}}^\ell(\boldsymbol{g}) \right| \right]$$

$$\leqslant \mathbb{E}_{\overline{\mathcal{D}}, \overline{\mathcal{D}}'} \left[ \sup_{g_1, g_2, \ldots, g_q \in \mathcal{G}} \left| \tilde{R}_{\mathrm{H}}^\ell(\boldsymbol{g}; \overline{\mathcal{D}}) - \tilde{R}_{\mathrm{H}}^\ell(\boldsymbol{g}'; \overline{\mathcal{D}}') \right| \right], \tag{22}$$

where the last inequality is deduced by applying Jensen's inequality twice. Here, $\tilde{R}_{\mathrm{H}}^{\ell}(\boldsymbol{g}; \widehat{\mathcal{D}})$ denotes the value of $\tilde{R}_{\mathrm{H}}^{\ell}(\boldsymbol{g})$ on $\widehat{\mathcal{D}}$. We introduce $\bar{\ell}(z) = \ell(z) - \ell(0)$ and we have $\bar{\ell}(z_1) - \bar{\ell}(z_2) = \ell(z_1) - \ell(z_2)$. It is obvious that $\bar{\ell}(z)$ is a Lipschitz continuous function with a Lipschitz constant $L_\ell$. Then, we have

$$
\begin{aligned}
&\left| \tilde{R}_{\mathrm{H}}^{\ell}(\boldsymbol{g}; \widehat{\mathcal{D}}) - \tilde{R}_{\mathrm{H}}^{\ell}(\boldsymbol{g}; \widehat{\mathcal{D}}') \right| \\
&\leqslant \frac{1}{q} \sum_{j=1}^{q} \left\| \left| \frac{1}{n} \sum_{i=1}^{n} \bar{\ell}\left(g_j\left(\boldsymbol{x}_i^{\mathrm{U}}\right), 1\right) - \frac{1-\pi_j}{n_j} \sum_{i=1}^{n_j} \bar{\ell}\left(g_j\left(\boldsymbol{x}_i^{j}\right), 1\right) \right| - \right. \\
&\qquad \left. \left| \frac{1}{n} \sum_{i=1}^{n} \bar{\ell}\left(g_j\left(\boldsymbol{x}_i^{\mathrm{U}'}\right), 1\right) - \frac{1-\pi_j}{n_j} \sum_{i=1}^{n_j} \bar{\ell}\left(g_j\left(\boldsymbol{x}_i^{j'}\right), 1\right) \right| \right\| \\
&\quad + \sum_{j=1}^{q} \left| \frac{1-\pi_j}{qn_j} \sum_{i=1}^{n_j} \bar{\ell}\left(g_j\left(\boldsymbol{x}_i^{j}\right), 0\right) - \frac{1-\pi_j}{qn_j} \sum_{i=1}^{n_j} \bar{\ell}\left(g_j\left(\boldsymbol{x}_i^{j'}\right), 0\right) \right| \\
&\leqslant \frac{1}{q} \sum_{j=1}^{q} \left| \frac{1}{n} \sum_{i=1}^{n} \bar{\ell}\left(g_j\left(\boldsymbol{x}_i^{\mathrm{U}}\right), 1\right) - \frac{1-\pi_j}{n_j} \sum_{i=1}^{n_j} \bar{\ell}\left(g_j\left(\boldsymbol{x}_i^{j}\right), 1\right) \right. \\
&\qquad \left. - \frac{1}{n} \sum_{i=1}^{n} \bar{\ell}\left(g_j\left(\boldsymbol{x}_i^{\mathrm{U}'}\right), 1\right) + \frac{1-\pi_j}{n_j} \sum_{i=1}^{n_j} \bar{\ell}\left(g_j\left(\boldsymbol{x}_i^{j'}\right), 1\right) \right| \\
&\quad + \sum_{j=1}^{q} \left| \frac{1-\pi_j}{qn_j} \sum_{i=1}^{n_j} \bar{\ell}\left(g_j\left(\boldsymbol{x}_i^{j}\right), 0\right) - \frac{1-\pi_j}{qn_j} \sum_{i=1}^{n_j} \bar{\ell}\left(g_j\left(\boldsymbol{x}_i^{j'}\right), 0\right) \right| \\
&\leqslant \frac{1}{q} \sum_{j=1}^{q} \left| \frac{1}{n} \sum_{i=1}^{n} \bar{\ell}\left(g_j\left(\boldsymbol{x}_i^{\mathrm{U}}\right), 1\right) - \frac{1}{n} \sum_{i=1}^{n} \bar{\ell}\left(g_j\left(\boldsymbol{x}_i^{\mathrm{U}'}\right), 1\right) \right| \\
&\quad + \frac{1}{q} \sum_{j=1}^{q} \left| \frac{1-\pi_j}{n_j} \sum_{i=1}^{n_j} \bar{\ell}\left(g_j\left(\boldsymbol{x}_i^{j'}\right), 1\right) - \frac{1-\pi_j}{n_j} \sum_{i=1}^{n_j} \bar{\ell}\left(g_j\left(\boldsymbol{x}_i^{j}\right), 1\right) \right| \\
&\quad + \sum_{j=1}^{q} \left| \frac{1-\pi_j}{qn_j} \sum_{i=1}^{n_j} \bar{\ell}\left(g_j\left(\boldsymbol{x}_i^{j}\right), 0\right) - \frac{1-\pi_j}{qn_j} \sum_{i=1}^{n_j} \bar{\ell}\left(g_j\left(\boldsymbol{x}_i^{j'}\right), 0\right) \right|,
\end{aligned}
\tag{23}
$$

where the inequalities are due to the triangle inequality. Then, by combining Inequalities (22) and (23), it is a routine work (Mohri et al., 2012) to show that

$$
\begin{aligned}
&\mathbb{E}_{\overline{\mathcal{D}}, \overline{\mathcal{D}}'} \left[ \sup_{g_1, g_2, \ldots, g_q \in \mathcal{G}} \left| \tilde{R}_{\mathrm{H}}^{\ell}(\boldsymbol{g}; \overline{\mathcal{D}}) - \tilde{R}_{\mathrm{H}}^{\ell}(\boldsymbol{g}'; \overline{\mathcal{D}}') \right| \right] \\
&\leqslant \frac{2}{q} \sum_{j=1}^{q} \mathfrak{R}'_{n,p}(\bar{\ell} \circ \mathcal{G}) + \frac{4(1-\pi_j)}{q} \sum_{j=1}^{q} \mathfrak{R}'_{n_j, p_j}(\bar{\ell} \circ \mathcal{G}) \\
&\leqslant \frac{4L_\ell}{q} \sum_{j=1}^{q} \mathfrak{R}'_{n,p}(\mathcal{G}) + \frac{8(1-\pi_j)L_\ell}{q} \sum_{j=1}^{q} \mathfrak{R}'_{n_j, p_j}(\mathcal{G}) \\
&= \frac{4L_\ell}{q} \sum_{j=1}^{q} \mathfrak{R}_{n,p}(\mathcal{G}) + \frac{8(1-\pi_j)L_\ell}{q} \sum_{j=1}^{q} \mathfrak{R}_{n_j, p_j}(\mathcal{G}),
\end{aligned}
\tag{24}
$$

where the second inequality is due to Lemma 4, $p_j$ denotes $p(\boldsymbol{x}|s_j = 0)$, and the last equality is due to Lemma 3. By combining Inequalities (21) and (24), we have the following inequality with probability at least $1 - \delta$:

$$
\begin{aligned}
&\sup_{g_1, g_2, \ldots, g_q \in \mathcal{G}} \left| \mathbb{E}\left[ \tilde{R}_{\mathrm{H}}^{\ell}(\boldsymbol{g}) \right] - \tilde{R}_{\mathrm{H}}^{\ell}(\boldsymbol{g}) \right| \leqslant \frac{4L_\ell}{q} \sum_{j=1}^{q} \mathfrak{R}_{n,p}(\mathcal{G}) + \frac{8(1-\pi_j)L_\ell}{q} \sum_{j=1}^{q} \mathfrak{R}_{n_j, p_j}(\mathcal{G}) \\
&\quad + C_\ell \sqrt{\frac{\ln(1/\delta)}{2n}} + \sum_{j=1}^{q} \frac{(2 - 2\pi_j)C_\ell}{q} \sqrt{\frac{\ln(1/\delta)}{2n_j}}.
\end{aligned}
\tag{25}
$$

Then, we have the following inequality with probability at least $1 - \delta$:

$$
\sup_{g_1, g_2, \ldots, g_q \in \mathcal{G}} \left| R_{\mathrm{H}}^{\ell}(\boldsymbol{g}) - \tilde{R}_{\mathrm{H}}^{\ell}(\boldsymbol{g}) \right|
$$

$$
= \sup_{g_1, g_2, \ldots, g_q \in \mathcal{G}} \left| R_{\mathrm{H}}^{\ell}(\boldsymbol{g}) - \mathbb{E}\left[ \tilde{R}_{\mathrm{H}}^{\ell}(\boldsymbol{g}) \right] + \mathbb{E}\left[ \tilde{R}_{\mathrm{H}}^{\ell}(\boldsymbol{g}) \right] - \tilde{R}_{\mathrm{H}}^{\ell}(\boldsymbol{g}) \right|
$$

$$
\leqslant \sup_{g_1, g_2, \ldots, g_q \in \mathcal{G}} \left| R_{\mathrm{H}}^{\ell}(\boldsymbol{g}) - \mathbb{E}\left[ \tilde{R}_{\mathrm{H}}^{\ell}(\boldsymbol{g}) \right] \right| + \sup_{g_1, g_2, \ldots, g_q \in \mathcal{G}} \left| \mathbb{E}\left[ \tilde{R}_{\mathrm{H}}^{\ell}(\boldsymbol{g}) \right] - \tilde{R}_{\mathrm{H}}^{\ell}(\boldsymbol{g}) \right|
$$

$$
\leqslant \frac{1}{q} \sum_{j=1}^{q} (4 - 2\pi_j) C_\ell \Delta_j + C_\ell \sqrt{\frac{\ln(1/\delta)}{2n}} + \sum_{j=1}^{q} \frac{(2 - 2\pi_j) C_\ell}{q} \sqrt{\frac{\ln(1/\delta)}{2n_j}}
$$

$$
+ \frac{4 L_\ell}{q} \sum_{j=1}^{q} \mathfrak{R}_{n,p}(\mathcal{G}) + \frac{8(1 - \pi_j) L_\ell}{q} \sum_{j=1}^{q} \mathfrak{R}_{n_j, p_j}(\mathcal{G}),
$$

where the second inequality is due to Inequalities 25 and 9. The proof is complete. $\qquad\square$

Then, we provide the proof of Theorem 3.

*Proof of Theorem 3.*

$$
R_{\mathrm{H}}^{\ell}(\tilde{\boldsymbol{g}}_{\mathrm{H}}) - R_{\mathrm{H}}^{\ell}(\boldsymbol{g}_{\mathrm{H}}^*) = R_{\mathrm{H}}^{\ell}(\tilde{\boldsymbol{g}}_{\mathrm{H}}) - \tilde{R}_{\mathrm{H}}^{\ell}(\tilde{\boldsymbol{g}}_{\mathrm{H}}) + \tilde{R}_{\mathrm{H}}^{\ell}(\tilde{\boldsymbol{g}}_{\mathrm{H}}) - \tilde{R}_{\mathrm{H}}^{\ell}(\boldsymbol{g}_{\mathrm{H}}^*) + \tilde{R}_{\mathrm{H}}^{\ell}(\boldsymbol{g}_{\mathrm{H}}^*) - R_{\mathrm{H}}^{\ell}(\boldsymbol{g}_{\mathrm{H}}^*)
$$

$$
\leqslant R_{\mathrm{H}}^{\ell}(\tilde{\boldsymbol{g}}_{\mathrm{H}}) - \tilde{R}_{\mathrm{H}}^{\ell}(\tilde{\boldsymbol{g}}_{\mathrm{H}}) + \tilde{R}_{\mathrm{H}}^{\ell}(\boldsymbol{g}_{\mathrm{H}}^*) - R_{\mathrm{H}}^{\ell}(\boldsymbol{g}_{\mathrm{H}}^*)
$$

$$
\leqslant 2 \sup_{g_1, g_2, \ldots, g_q \in \mathcal{G}} \left| R_{\mathrm{H}}^{\ell}(\boldsymbol{g}) - \tilde{R}_{\mathrm{H}}^{\ell}(\boldsymbol{g}) \right|.
$$

By Lemma 5, the proof is complete. $\qquad\square$

### B.5 PROOF OF COROLLARY 1

**Lemma 6** (Theorem 4 in Gao & Zhou (2013)). *The surrogate loss $R^\ell$ is multi-label consistent w.r.t. the Hamming or ranking loss $R^{0-1}$ if and only if it holds for any sequence $\{\boldsymbol{g}_t\}$ that if $R^\ell(\boldsymbol{g}) \to R^{\ell*}$, then $R^{0-1}(\boldsymbol{g}) \to R^*$. Here, $R^{\ell*} = \inf_{\boldsymbol{g}} R^\ell(\boldsymbol{g})$ and $R^* = \inf_{\boldsymbol{g}} R^{0-1}(\boldsymbol{g})$.*

**Lemma 7** (Theorem 32 in Gao & Zhou (2013)). *If $\ell$ is a convex function with $\ell'(0, y) < 0$, then Eq. (2) is consistent w.r.t. the Hamming loss.*

Then, we provide the proof of Corollary 1.

*Proof of Corollary 1.* Since the proposed risk in Eq. (5) is equivalent to the risk in Eq. (2), it is sufficient to prove that for any sequence $\{\boldsymbol{g}_t\}$ that if $R_{\mathrm{H}}^{\ell}(\boldsymbol{g}_t) \to R_{\mathrm{H}}^{\ell*}$, then $R_{\mathrm{H}}^{0-1}(\boldsymbol{f}_t) \to R_{\mathrm{H}}^*$. $\qquad\square$

## B.6 PROOF OF THEOREM 4

$$
\begin{aligned}
R_{\mathrm{R}}^{\ell}(\boldsymbol{g}) =& \mathbb{E}_{p(\boldsymbol{x},Y)} \left[ \sum_{1 \leqslant j < k \leqslant q} \mathbb{I}(y_j \neq y_k) \ell \left( g_j(\boldsymbol{x}) - g_k(\boldsymbol{x}), \frac{y_j - y_k + 1}{2} \right) \right] \\
=& \int \sum_Y \sum_{1 \leqslant j < k \leqslant q} \mathbb{I}(y_j \neq y_k) \ell \left( g_j(\boldsymbol{x}) - g_k(\boldsymbol{x}), \frac{y_j - y_k + 1}{2} \right) p(\boldsymbol{x}, Y) \, \mathrm{d}\boldsymbol{x} \\
=& \int \sum_{1 \leqslant j < k \leqslant q} \sum_{y_j=0}^{1} \sum_{y_k=0}^{1} \sum_{Y'=Y \backslash \{y_j,y_k\}} \mathbb{I}(y_j \neq y_k) \ell \left( g_j(\boldsymbol{x}) - g_k(\boldsymbol{x}), \frac{y_j - y_k + 1}{2} \right) \\
& p(\boldsymbol{x}, y_j, y_k) \, p\left(Y'|\boldsymbol{x}, y_j, y_k\right) \, \mathrm{d}\boldsymbol{x} \\
=& \int \sum_{1 \leqslant j < k \leqslant q} \sum_{y_j=0}^{1} \sum_{y_k=0}^{1} \mathbb{I}(y_j \neq y_k) \ell \left( g_j(\boldsymbol{x}) - g_k(\boldsymbol{x}), \frac{y_j - y_k + 1}{2} \right) \\
& p(\boldsymbol{x}, y_j, y_k) \sum_{Y'=Y \backslash \{y_j,y_k\}} p\left(Y'|\boldsymbol{x}, y_j, y_k\right) \, \mathrm{d}\boldsymbol{x} \\
=& \int \sum_{1 \leqslant j < k \leqslant q} \sum_{y_j=0}^{1} \sum_{y_k=0}^{1} \mathbb{I}(y_j \neq y_k) \ell \left( g_j(\boldsymbol{x}) - g_k(\boldsymbol{x}), \frac{y_j - y_k + 1}{2} \right) p(\boldsymbol{x}, y_j, y_k) \, \mathrm{d}\boldsymbol{x} \\
=& \sum_{1 \leqslant j < k \leqslant q} \left( \int \ell\left(g_j(\boldsymbol{x}) - g_k(\boldsymbol{x}), 0\right) p(\boldsymbol{x}, y_j = 0, y_k = 1) \, \mathrm{d}\boldsymbol{x} \right. \\
& \left. + \int \ell\left(g_j(\boldsymbol{x}) - g_k(\boldsymbol{x}), 1\right) p(\boldsymbol{x}, y_j = 1, y_k = 0) \, \mathrm{d}\boldsymbol{x} \right) \\
=& \sum_{1 \leqslant j < k \leqslant q} \left( \int \ell\left(g_j(\boldsymbol{x}) - g_k(\boldsymbol{x}), 0\right) \left(p(\boldsymbol{x}, y_j = 0) - p(\boldsymbol{x}, y_j = 0, y_k = 0)\right) \, \mathrm{d}\boldsymbol{x} \right. \\
& \left. + \int \ell\left(g_j(\boldsymbol{x}) - g_k(\boldsymbol{x}), 1\right) \left(p(\boldsymbol{x}, y_k = 0) - p(\boldsymbol{x}, y_j = 0, y_k = 0)\right) \, \mathrm{d}\boldsymbol{x} \right) \\
=& \sum_{1 \leqslant j < k \leqslant q} \left( \int \ell\left(g_j(\boldsymbol{x}) - g_k(\boldsymbol{x}), 0\right) (1 - \pi_j) p(\boldsymbol{x}|y_j = 0) \, \mathrm{d}\boldsymbol{x} \right. \\
& + \int \ell\left(g_j(\boldsymbol{x}) - g_k(\boldsymbol{x}), 1\right) (1 - \pi_k) p(\boldsymbol{x}|y_k = 0) \, \mathrm{d}\boldsymbol{x} \\
& \left. - \int \left(\ell\left(g_j(\boldsymbol{x}) - g_k(\boldsymbol{x}), 0\right) + \ell\left(g_j(\boldsymbol{x}) - g_k(\boldsymbol{x}), 1\right)\right) p(\boldsymbol{x}, y_j = 0, y_k = 0) \, \mathrm{d}\boldsymbol{x} \right) \\
=& \sum_{1 \leqslant j < k \leqslant q} \left( (1 - \pi_j) \mathbb{E}_{p(\boldsymbol{x}|y_j=0)} \left[ \ell\left(g_j(\boldsymbol{x}) - g_k(\boldsymbol{x}), 0\right) \right] \right. \\
& \left. + (1 - \pi_k) \mathbb{E}_{p(\boldsymbol{x}|y_k=0)} \left[ \ell\left(g_j(\boldsymbol{x}) - g_k(\boldsymbol{x}), 1\right) \right] - M p(y_j = 0, y_k = 0) \right). \\
=& \sum_{1 \leqslant j < k \leqslant q} \left( (1 - \pi_j) \mathbb{E}_{p(\boldsymbol{x}|s_j=0)} \left[ \ell\left(g_j(\boldsymbol{x}) - g_k(\boldsymbol{x}), 0\right) \right] \right. \\
& \left. + (1 - \pi_k) \mathbb{E}_{p(\boldsymbol{x}|s_k=0)} \left[ \ell\left(g_j(\boldsymbol{x}) - g_k(\boldsymbol{x}), 1\right) \right] - M p(y_j = 0, y_k = 0) \right),
\end{aligned}
$$

where the last equation is by Lemma 1. The proof is completed. $\qquad\square$

## B.7 PROOF OF THEOREM 5

Let $\widehat{\mathcal{D}} = \mathcal{D}_1 \bigcup \mathcal{D}_2 \bigcup \ldots \mathcal{D}_q$ denote the set of all the data used in Eq. (14). We introduce

$$
\mathfrak{D}^+(\boldsymbol{g}) = \left\{ \widehat{\mathcal{D}} | \hat{R}_{\mathrm{R}}^{\ell}(\boldsymbol{g}) > \beta \right\}, \quad \text{and} \quad \mathfrak{D}^-(\boldsymbol{g}) = \left\{ \widehat{\mathcal{D}} | \hat{R}_{\mathrm{R}}^{\ell}(\boldsymbol{g}) \leqslant \beta \right\}.
$$

Then, we have the following lemma.

**Lemma 8.** *The probability measure of $\mathfrak{D}^-(\boldsymbol{g})$ can be bounded as follows:*

$$
\mathbb{P}\left(\mathfrak{D}^-(\boldsymbol{g})\right) \leqslant \exp\left( \frac{-2\gamma^2}{\sum_{j=1}^{q} (1 - \pi_j)^2 (q-1)^2 C_\ell^2 / n_j} \right). \tag{26}
$$

*Proof.* When an instance from $\mathcal{D}_j$ is replaced by another instance, the value of $\hat{R}_{\mathrm{R}}^\ell(\boldsymbol{g})$ changes at most $(1 - \pi_j)(q - 1)C_\ell/n_j$. Therefore, by applying the McDiarmid's inequality, we can obtain the following inequality:

$$p\left( R_{\mathrm{R}}^\ell(\boldsymbol{g}) - \hat{R}_{\mathrm{R}}^\ell(\boldsymbol{g}) + \sum_{1 \leqslant j < k \leqslant q} Mp(y_j = 0, y_k = 0) \geqslant \gamma \right) \leqslant \exp\left( \frac{-2\gamma^2}{\sum_{j=1}^q (1 - \pi_j)^2(q - 1)^2 C_\ell^2/n_j} \right).$$

Then, we have

$$\begin{aligned}
\mathbb{P}\left( \mathfrak{D}^-(\boldsymbol{g}) \right) &= p\left( \hat{R}_{\mathrm{R}}^\ell(\boldsymbol{g}) \leqslant \beta \right) \\
&\leqslant p\left( \hat{R}_{\mathrm{R}}^\ell(\boldsymbol{g}) \leqslant \sum_{1 \leqslant j < k \leqslant q} Mp(y_j = 0, y_k = 0) \right) \\
&\leqslant p\left( \hat{R}_{\mathrm{R}}^\ell(\boldsymbol{g}) \leqslant \sum_{1 \leqslant j < k \leqslant q} Mp(y_j = 0, y_k = 0) + R_{\mathrm{R}}^\ell(\boldsymbol{g}) - \gamma \right) \\
&\leqslant \exp\left( \frac{-2\gamma^2}{\sum_{j=1}^q (1 - \pi_j)^2(q - 1)^2 C_\ell^2/n_j} \right),
\end{aligned}$$

which concludes the proof. $\square$

Then, we give the proof of Theorem 5.

*Proof of Theorem 5.* To begin with, we have

$$\mathbb{E}\left[ \tilde{R}_{\mathrm{R}}^\ell(\boldsymbol{g}) \right] - \sum_{1 \leqslant j < k \leqslant q} Mp(y_j = 0, y_k = 0) - R_{\mathrm{R}}^\ell(\boldsymbol{g}) = \mathbb{E}\left[ \tilde{R}_{\mathrm{R}}^\ell(\boldsymbol{g}) - \hat{R}_{\mathrm{R}}^\ell(\boldsymbol{g}) \right] \geqslant 0.$$

Besides, we have

$$\hat{R}_{\mathrm{R}}^\ell(\boldsymbol{g}) \leqslant C_\ell(q - 1) \sum_{j=1}^q (1 - \pi_j).$$

Then,

$$\begin{aligned}
&\mathbb{E}\left[ \tilde{R}_{\mathrm{R}}^\ell(\boldsymbol{g}) \right] - \sum_{1 \leqslant j < k \leqslant q} Mp(y_j = 0, y_k = 0) - R_{\mathrm{R}}^\ell(\boldsymbol{g}) \\
=&\mathbb{E}\left[ \tilde{R}_{\mathrm{R}}^\ell(\boldsymbol{g}) - \hat{R}_{\mathrm{R}}^\ell(\boldsymbol{g}) \right] \\
=&\int_{\hat{\mathcal{D}} \in \mathfrak{D}^-(\boldsymbol{g})} \left( \left| \hat{R}_{\mathrm{R}}^\ell(\boldsymbol{g}) - \beta \right| + \beta - \hat{R}_{\mathrm{R}}^\ell(\boldsymbol{g}) \right) p\left( \hat{\mathcal{D}} \right) \mathrm{d}\hat{\mathcal{D}} \\
\leqslant&\sup_{\hat{\mathcal{D}} \in \mathfrak{D}^-(\boldsymbol{g})} \left( 2\beta + 2\hat{R}_{\mathrm{R}}^\ell(\boldsymbol{g}) \right) \int_{\hat{\mathcal{D}} \in \mathfrak{D}^-(\boldsymbol{g})} p\left( \hat{\mathcal{D}} \right) \mathrm{d}\hat{\mathcal{D}} \\
=&\sup_{\hat{\mathcal{D}} \in \mathfrak{D}^-(\boldsymbol{g})} \left( 2\beta + 2\hat{R}_{\mathrm{R}}^\ell(\boldsymbol{g}) \right) \mathbb{P}\left( \mathfrak{D}^-(\boldsymbol{g}) \right) \\
\leqslant&\left( 2\beta + 2C_\ell(q - 1) \sum_{j=1}^q (1 - \pi_j) \right) \exp\left( \frac{-2\gamma^2}{\sum_{j=1}^q (1 - \pi_j)^2(q - 1)^2 C_\ell^2/n_j} \right),
\end{aligned}$$

which concludes the first part of the proof. Then we provide an upper bound for $\left| \tilde{R}_{\mathrm{R}}^\ell(\boldsymbol{g}) - \mathbb{E}\left[ \tilde{R}_{\mathrm{R}}^\ell(\boldsymbol{g}) \right] \right|$. When an instance from $\mathcal{D}_j$ is replaced by another instance, the value of $\tilde{R}_{\mathrm{R}}^\ell(\boldsymbol{g})$ changes at most $(1 - \pi_j)(q - 1)C_\ell/n_j$. Therefore, by applying the McDiarmid's inequality, we have the following inequalities with probability at least $1 - \delta/2$:

$$\tilde{R}_{\mathrm{R}}^\ell(\boldsymbol{g}) - \mathbb{E}\left[ \tilde{R}_{\mathrm{R}}^\ell(\boldsymbol{g}) \right] \leqslant \sum_{j=1}^q (1 - \pi_j)(q - 1)C_\ell \sqrt{\frac{\ln(2/\delta)}{2n_j}},$$

$$\mathbb{E}\left[ \tilde{R}_{\mathrm{R}}^\ell(\boldsymbol{g}) \right] - \tilde{R}_{\mathrm{R}}^\ell(\boldsymbol{g}) \leqslant \sum_{j=1}^q (1 - \pi_j)(q - 1)C_\ell \sqrt{\frac{\ln(2/\delta)}{2n_j}}.$$

Therefore, we have the following inequalities with probability at least $1 - \delta$:

$$\left| \tilde{R}_{\mathrm{R}}^{\ell}(\boldsymbol{g}) - \mathbb{E}\left[ \tilde{R}_{\mathrm{R}}^{\ell}(\boldsymbol{g}) \right] \right| \leqslant \sum_{j=1}^{q} (1 - \pi_j)(q-1)C_{\ell}\sqrt{\frac{\ln(2/\delta)}{2n_j}}.$$

Finally,

$$
\begin{aligned}
&\left| \tilde{R}_{\mathrm{R}}^{\ell}(\boldsymbol{g}) - \sum_{1 \leqslant j < k \leqslant q} M p(y_j = 0, y_k = 0) - R_{\mathrm{R}}^{\ell}(\boldsymbol{g}) \right| \\
&= \left| \tilde{R}_{\mathrm{R}}^{\ell}(\boldsymbol{g}) - \mathbb{E}\left[ \tilde{R}_{\mathrm{R}}^{\ell}(\boldsymbol{g}) \right] + \mathbb{E}\left[ \tilde{R}_{\mathrm{R}}^{\ell}(\boldsymbol{g}) \right] - \sum_{1 \leqslant j < k \leqslant q} M p(y_j = 0, y_k = 0) - R_{\mathrm{R}}^{\ell}(\boldsymbol{g}) \right| \\
&\leqslant \left| \tilde{R}_{\mathrm{R}}^{\ell}(\boldsymbol{g}) - \mathbb{E}\left[ \tilde{R}_{\mathrm{R}}^{\ell}(\boldsymbol{g}) \right] \right| + \left| \mathbb{E}\left[ \tilde{R}_{\mathrm{R}}^{\ell}(\boldsymbol{g}) \right] - \sum_{1 \leqslant j < k \leqslant q} M p(y_j = 0, y_k = 0) - R_{\mathrm{R}}^{\ell}(\boldsymbol{g}) \right| \\
&\leqslant \sum_{j=1}^{q} (1 - \pi_j)(q-1)C_{\ell}\sqrt{\frac{\ln(2/\delta)}{2n_j}} \\
&\quad + \left( 2\beta + 2C_{\ell}(q-1)\sum_{j=1}^{q}(1 - \pi_j) \right) \exp\left( \frac{-2\gamma^2}{\sum_{j=1}^{q}(1 - \pi_j)^2(q-1)^2 C_{\ell}^2/n_j} \right),
\end{aligned}
\tag{27}
$$

which concludes the proof. $\qquad\square$

### B.8 PROOF OF THEOREM 6

**Lemma 9.** *Based on the above assumptions, for any $\delta > 0$, the following inequality holds with probability at least $1 - \delta$:*

$$
\sup_{g_1, g_2, \ldots, g_q \in \mathcal{G}} \left| R_{\mathrm{R}}^{\ell}(\boldsymbol{g}) + \sum_{j < k} M p(y_j = 0, y_k = 0) - \tilde{R}_{\mathrm{R}}^{\ell}(\boldsymbol{g}) \right| \leqslant \left( 2\beta + 2C_{\ell}(q-1)\sum_{j=1}^{q}(1 - \pi_j) \right)\Delta'
$$

$$
+ \sum_{j=1}^{q}(1 - \pi_j)(q-1)C_{\ell}\sqrt{\frac{\ln(1/\delta)}{n_j}} + \sum_{j=1}^{q} 4L_{\ell}(q-1)(1 - \pi_j)\mathfrak{R}_{n_j, p_j}(\mathcal{G}).
\tag{28}
$$

*Proof.* When an instance in $\mathcal{D}_j$ is replaced by another instance, the value of $\sup_{g_1, g_2, \ldots, g_q \in \mathcal{G}} \left| \mathbb{E}\left[ \tilde{R}_{\mathrm{R}}^{\ell}(\boldsymbol{g}) \right] - \tilde{R}_{\mathrm{R}}^{\ell}(\boldsymbol{g}) \right|$ changes at most $(1 - \pi_j)(q-1)C_{\ell}/n_j$. Therefore, by applying the McDiarmid's inequality, we have the following inequalities with probability at least $1 - \delta$:

$$
\begin{aligned}
&\sup_{g_1, g_2, \ldots, g_q \in \mathcal{G}} \left| \mathbb{E}\left[ \tilde{R}_{\mathrm{R}}^{\ell}(\boldsymbol{g}) \right] - \tilde{R}_{\mathrm{R}}^{\ell}(\boldsymbol{g}) \right| - \mathbb{E}\left[ \sup_{g_1, g_2, \ldots, g_q \in \mathcal{G}} \left| \mathbb{E}\left[ \tilde{R}_{\mathrm{R}}^{\ell}(\boldsymbol{g}) \right] - \tilde{R}_{\mathrm{R}}^{\ell}(\boldsymbol{g}) \right| \right] \\
&\leqslant \sum_{j=1}^{q}(1 - \pi_j)(q-1)C_{\ell}\sqrt{\frac{\ln(1/\delta)}{n_j}}.
\end{aligned}
\tag{29}
$$

Then,

$$
\begin{aligned}
&\mathbb{E}\left[ \sup_{g_1, g_2, \ldots, g_q \in \mathcal{G}} \left| \mathbb{E}\left[ \tilde{R}_{\mathrm{R}}^{\ell}(\boldsymbol{g}) \right] - \tilde{R}_{\mathrm{R}}^{\ell}(\boldsymbol{g}) \right| \right] \\
&= \mathbb{E}_{\widehat{\mathcal{D}}}\left[ \sup_{g_1, g_2, \ldots, g_q \in \mathcal{G}} \left| \mathbb{E}_{\widehat{\mathcal{D}}'}\left[ \tilde{R}_{\mathrm{R}}^{\ell}(\boldsymbol{g}) \right] - \tilde{R}_{\mathrm{R}}^{\ell}(\boldsymbol{g}) \right| \right] \\
&\leqslant \mathbb{E}_{\widehat{\mathcal{D}}, \widehat{\mathcal{D}}'}\left[ \sup_{g_1, g_2, \ldots, g_q \in \mathcal{G}} \left| \tilde{R}_{\mathrm{R}}^{\ell}(\boldsymbol{g}; \widehat{\mathcal{D}}) - \tilde{R}_{\mathrm{R}}^{\ell}(\boldsymbol{g}'; \widehat{\mathcal{D}}') \right| \right],
\end{aligned}
\tag{30}
$$

where the last inequality is deduced by applying Jensen's inequality twice. Here, $\tilde{R}_{\mathrm{R}}^{\ell}(\boldsymbol{g}; \widehat{\mathcal{D}})$ denotes the value of $\tilde{R}_{\mathrm{R}}^{\ell}(\boldsymbol{g})$ on $\widehat{\mathcal{D}}$. Then, we introduce the following lemma.

**Lemma 10.** *If $\ell : \mathbb{R} \times \{0, 1\} \to \mathbb{R}$ is a Lipschitz continuous function with a Lipschitz constant $L_\ell$ and satisfies $\forall y, \ell(0, y) = 0$, we have*

$$\mathfrak{R}'_{n,p}\left(\ell \circ (\mathcal{G} - \mathcal{G})\right) \leqslant 4 L_\ell \mathfrak{R}'_{n,p}(\mathcal{G}),$$

*where $\ell \circ ((\mathcal{G} - \mathcal{G})) = \{\ell \circ (g_j - g_k) \mid g_j \in \mathcal{G}, g_k \in \mathcal{G}\}$.*

*Proof.*

$$
\begin{aligned}
&\mathfrak{R}'_{n,p}\left(\ell \circ (\mathcal{G} - \mathcal{G})\right) \\
=& 2\mathfrak{R}'_{n,p}\left(\ell \circ (\mathcal{G})\right) \\
\leqslant& 4 L_\ell \mathfrak{R}'_{n,p}(\mathcal{G}),
\end{aligned}
$$

where the first inequality is by symmetrization (Mohri et al., 2012) and the second inequality is by Lemma 4. The proof is complete. □

Therefore, we have

$$
\begin{aligned}
&\left| \tilde{R}_{\mathrm{R}}^\ell(\boldsymbol{g}; \widehat{\mathcal{D}}) - \tilde{R}_{\mathrm{R}}^\ell(\boldsymbol{g}; \widehat{\mathcal{D}}') \right| \\
=& \left| \left| \hat{R}_{\mathrm{R}}^\ell(\boldsymbol{g}; \widehat{\mathcal{D}}) - \beta \right| - \left| \hat{R}_{\mathrm{R}}^\ell(\boldsymbol{g}; \widehat{\mathcal{D}}') - \beta \right| \right| \\
\leqslant& \left| \hat{R}_{\mathrm{R}}^\ell(\boldsymbol{g}; \widehat{\mathcal{D}}) - \hat{R}_{\mathrm{R}}^\ell(\boldsymbol{g}; \widehat{\mathcal{D}}') \right| \\
=& \left| \sum_{1 \leqslant j < k \leqslant q} \frac{1 - \pi_j}{n_j} \sum_{i=1}^{n_j} \left( \ell\left(g_j(\boldsymbol{x}_i^j) - g_k(\boldsymbol{x}_i^j), 0\right) - \ell\left(g_j(\boldsymbol{x}_i^{j'}) - g_k(\boldsymbol{x}_i^{j'}), 0\right) \right) \right. \\
&\left. + \frac{1 - \pi_k}{n_k} \sum_{i=1}^{n_k} \left( \ell\left(g_j(\boldsymbol{x}_i^k) - g_k(\boldsymbol{x}_i^k), 1\right) - \ell\left(g_j(\boldsymbol{x}_i^{k'}) - g_k(\boldsymbol{x}_i^{k'}), 1\right) \right) \right| \\
\leqslant& \sum_{1 \leqslant j < k \leqslant q} \left| \frac{1 - \pi_j}{n_j} \sum_{i=1}^{n_j} \left( \ell\left(g_j(\boldsymbol{x}_i^j) - g_k(\boldsymbol{x}_i^j), 0\right) - \ell\left(g_j(\boldsymbol{x}_i^{j'}) - g_k(\boldsymbol{x}_i^{j'}), 0\right) \right) \right| \\
&+ \sum_{1 \leqslant j < k \leqslant q} \left| \frac{1 - \pi_k}{n_k} \sum_{i=1}^{n_k} \left( \ell\left(g_j(\boldsymbol{x}_i^k) - g_k(\boldsymbol{x}_i^k), 1\right) - \ell\left(g_j(\boldsymbol{x}_i^{k'}) - g_k(\boldsymbol{x}_i^{k'}), 1\right) \right) \right| \\
=& \sum_{1 \leqslant j < k \leqslant q} \left| \frac{1 - \pi_j}{n_j} \sum_{i=1}^{n_j} \left( \bar{\ell}\left(g_j(\boldsymbol{x}_i^j) - g_k(\boldsymbol{x}_i^j), 0\right) - \bar{\ell}\left(g_j(\boldsymbol{x}_i^{j'}) - g_k(\boldsymbol{x}_i^{j'}), 0\right) \right) \right| \\
&+ \sum_{1 \leqslant j < k \leqslant q} \left| \frac{1 - \pi_k}{n_k} \sum_{i=1}^{n_k} \left( \bar{\ell}\left(g_j(\boldsymbol{x}_i^k) - g_k(\boldsymbol{x}_i^k), 1\right) - \bar{\ell}\left(g_j(\boldsymbol{x}_i^{k'}) - g_k(\boldsymbol{x}_i^{k'}), 1\right) \right) \right|, \quad (31)
\end{aligned}
$$

where the inequalities are due to the triangle inequality. Then, by combining Inequalities 30 and 31, it is a routine work (Mohri et al., 2012) to show that

$$
\begin{aligned}
&\mathbb{E}\left[ \sup_{g_1, g_2, \ldots, g_q \in \mathcal{G}} \left| \mathbb{E}\left[ \tilde{R}_{\mathrm{R}}^\ell(\boldsymbol{g}) \right] - \tilde{R}_{\mathrm{R}}^\ell(\boldsymbol{g}) \right| \right] \\
\leqslant& \sum_{j=1}^{q} (q - 1)(1 - \pi_j)\mathfrak{R}'_{n_j, p_j}\left( \bar{\ell} \circ (\mathcal{G} - \mathcal{G}) \right) \\
\leqslant& \sum_{j=1}^{q} 4 L_\ell (q - 1)(1 - \pi_j)\mathfrak{R}'_{n_j, p_j}(\mathcal{G}) \\
=& \sum_{j=1}^{q} 4 L_\ell (q - 1)(1 - \pi_j)\mathfrak{R}_{n_j, p_j}(\mathcal{G}), \quad (32)
\end{aligned}
$$

where the second inequality is by Lemma 10 and the last equality is by Lemma 3. By combining Inequalities 29 and 32, we have the following inequalities with probability at least $1 - \delta$:

$$
\begin{aligned}
&\sup_{g_1, g_2, \ldots, g_q \in \mathcal{G}} \left| \mathbb{E}\left[ \tilde{R}_{\mathrm{R}}^\ell(\boldsymbol{g}) \right] - \tilde{R}_{\mathrm{R}}^\ell(\boldsymbol{g}) \right| \\
&\leqslant \sum_{j=1}^{q} (1 - \pi_j)(q - 1)C_\ell \sqrt{\frac{\ln(1/\delta)}{n_j}} + \sum_{j=1}^{q} 4 L_\ell (q - 1)(1 - \pi_j)\mathfrak{R}_{n_j, p_j}(\mathcal{G}). \quad (33)
\end{aligned}
$$

Finally, we have the following inequality with probability at least $1 - \delta$:

$$\sup_{g_1, g_2, \ldots, g_q \in \mathcal{G}} \left| R_{\mathrm{R}}^\ell(\boldsymbol{g}) + \sum_{j < k} M p(y_j = 0, y_k = 0) - \tilde{R}_{\mathrm{R}}^\ell(\boldsymbol{g}) \right|$$

$$= \sup_{g_1, g_2, \ldots, g_q \in \mathcal{G}} \left| R_{\mathrm{R}}^\ell(\boldsymbol{g}) + \sum_{j < k} M p(y_j = 0, y_k = 0) - \mathbb{E}\left[\tilde{R}_{\mathrm{R}}^\ell(\boldsymbol{g})\right] + \mathbb{E}\left[\tilde{R}_{\mathrm{R}}^\ell(\boldsymbol{g})\right] - \tilde{R}_{\mathrm{R}}^\ell(\boldsymbol{g}) \right|$$

$$\leqslant \sup_{g_1, g_2, \ldots, g_q \in \mathcal{G}} \left| R_{\mathrm{R}}^\ell(\boldsymbol{g}) + \sum_{j < k} M p(y_j = 0, y_k = 0) - \mathbb{E}\left[\tilde{R}_{\mathrm{R}}^\ell(\boldsymbol{g})\right] \right| + \sup_{g_1, g_2, \ldots, g_q \in \mathcal{G}} \left| \mathbb{E}\left[\tilde{R}_{\mathrm{R}}^\ell(\boldsymbol{g})\right] - \tilde{R}_{\mathrm{R}}^\ell(\boldsymbol{g}) \right|$$

$$\leqslant \left(2\beta + 2C_\ell(q - 1) \sum_{j=1}^q (1 - \pi_j)\right) \Delta' + \sum_{j=1}^q (1 - \pi_j)(q - 1) C_\ell \sqrt{\frac{\ln(1/\delta)}{n_j}}$$

$$+ \sum_{j=1}^q 4 L_\ell (q - 1)(1 - \pi_j) \mathfrak{R}_{n_j, p_j}(\mathcal{G}),$$

where the last inequality is by Inequalities 33 and 15. The proof is complete. □

Then, we provide the proof of Theorem 6.

*Proof of Theorem 6.*

$$R_{\mathrm{R}}^\ell(\tilde{\boldsymbol{g}}_{\mathrm{R}}) - R_{\mathrm{R}}^\ell(\boldsymbol{g}_{\mathrm{R}}^*)$$

$$= R_{\mathrm{R}}^\ell(\tilde{\boldsymbol{g}}_{\mathrm{R}}) + \sum_{j < k} M p(y_j = 0, y_k = 0) - \tilde{R}_{\mathrm{R}}^\ell(\tilde{\boldsymbol{g}}_{\mathrm{R}}) + \tilde{R}_{\mathrm{R}}^\ell(\tilde{\boldsymbol{g}}_{\mathrm{R}}) - \tilde{R}_{\mathrm{R}}^\ell(\boldsymbol{g}_{\mathrm{R}}^*) + \tilde{R}_{\mathrm{R}}^\ell(\boldsymbol{g}_{\mathrm{R}}^*)$$

$$- \sum_{j < k} M p(y_j = 0, y_k = 0) - R_{\mathrm{R}}^\ell(\boldsymbol{g}_{\mathrm{R}}^*)$$

$$\leqslant R_{\mathrm{R}}^\ell(\tilde{\boldsymbol{g}}_{\mathrm{R}}) + \sum_{j < k} M p(y_j = 0, y_k = 0) - \tilde{R}_{\mathrm{R}}^\ell(\tilde{\boldsymbol{g}}_{\mathrm{R}}) + \tilde{R}_{\mathrm{R}}^\ell(\boldsymbol{g}_{\mathrm{R}}^*) - \sum_{j < k} M p(y_j = 0, y_k = 0) - R_{\mathrm{R}}^\ell(\boldsymbol{g}_{\mathrm{R}}^*)$$

$$\leqslant 2 \sup_{g_1, g_2, \ldots, g_q \in \mathcal{G}} \left| R_{\mathrm{R}}^\ell(\boldsymbol{g}) + \sum_{j < k} M p(y_j = 0, y_k = 0) - \tilde{R}_{\mathrm{R}}^\ell(\boldsymbol{g}) \right|.$$

Then, based on Lemma 9, the proof is complete. □

## B.9 PROOF OF COROLLARY 2

**Lemma 11** (Theorem 10 in Gao & Zhou (2013))**.** *If $\ell$ is a differentiable and non-increasing function such that $\forall y, \ell'(0, y) < 0$ and $\ell(z, y) + \ell(-z, y) = M$, then Eq. (4) is consistent w.r.t. the ranking loss.*

Then we provide the proof of Corollary 2.

*Proof of Corollary 2.* Since the proposed risk in Eq. (5) is equivalent to the risk in Eq. (4), it is sufficient to prove that for any sequence $\{\boldsymbol{g}_t\}$ that if $R_{\mathrm{R}}^\ell(\boldsymbol{g}_t) \to R_{\mathrm{R}}^{\ell*}$, then $R_{\mathrm{R}}^{0\text{-}1}(\boldsymbol{f}_t) \to R_{\mathrm{R}}^*$. □

## C DETAILS OF EXPERIMENTS

### C.1 MORE DETAILS OF DATASETS

For synthetic datasets, we consider two data generation processes. In case-a, irrelevant labels are flipped to candidate labels independently, which is the assumption used in Xie & Huang (2023). This strategy is common in learning with noisy labels (Han et al., 2018), where PML is a special case of MLC with noisy labels (Xie & Huang, 2023). In case-b, we assign non-candidate labels in a class-wise manner. For each class, we randomly sample a fraction of the training data and assign that class as a non-candidate label. This data generation process corresponds to the assumption proposed in this paper. We use this process to confirm the effectiveness of our proposed method under this assumption. Additionally, we selected high flipping rates to evaluate the effectiveness of our proposed methods on challenging datasets with high noise rates since real-world datasets have low noise rates. In this paper, we set the flipping rate in Case-a to be 0.9 and and the sampling rate in Case-b to be 0.1.

## C.2 BASELINES

We evaluate against five classical baselines commonly used in PML/CML learning. (A) BCE: uses the given candidate label as the cross-entropy target. (B) CCMN (Xie & Huang, 2023): treats PML as multi-label classification with class-conditional noise, relying on a noise transition matrix. (C) GDF (Gao et al., 2023): proposes an unbiased risk estimator for multi-labeled single complementary label learning. (D) CTL (Gao et al., 2025): introduces a risk-consistent approach by rewriting the loss function. (E) MLCL (Gao et al., 2024): estimates an initial transition matrix via binary decompositions, then refines it with label correlations.

# D INSTANCE-DEPENDENT CASES

The current literature on partial multi-label learning (PML) and complementary multi-label learning (CML) assumes that label generation is independent of instances (see Table 1). Following previous work, we also consider the instance-independent case. It is very challenging to design consistent methods for instance-dependent cases due to the difficulty of estimating instance-dependent generation processes, as far as we know from the literature on weakly supervised learning. In future work, we will consider developing instance-dependent methods with strong theoretical guarantees.

