# OpenReview forum: "Rethinking Consistent Multi-Label Classification Under Inexact Supervision"
_ICLR.cc/2026/Conference — ICLR 2026 Poster_

### Official Review · Reviewer_2V1q · 2025-10-28

**Soundness:** 3
**Presentation:** 3
**Contribution:** 3
**Rating:** 6
**Confidence:** 4

**Summary:**

This paper presents a unified framework, COMES, for consistent multi-label classification under inexact supervision, targeting both Partial Multi-Label (PML) and Complementary Multi-Label (CML) learning. The authors aim to overcome the limitations of existing methods, which rely on either estimating the complex label generation process or adopting a uniform distribution assumption. The core contribution is a new data generation assumption—that true negatives are marked as non-candidates with a constant, instance-independent probability—which allows for a new approach. Based on this premise, the paper derives unbiased risk estimators (and their subsequent corrected, consistent versions) for two widely used metrics: the Hamming loss (COMES-HL, a first-order strategy) and the Ranking loss (COMES-RL, a second-order strategy). The authors provide theoretical guarantees for the consistency and estimation error bounds of their proposed estimators. Empirically, the framework is validated on ten benchmark datasets, where it is shown to outperform current state-of-the-art methods.

**Strengths:**

1.  The paper is well-structured and clearly articulates its approach (COMES) as a unified solution for both PML and CML problems.
2.  It introduces a new data generation assumption that avoids the common pitfalls of transition matrix estimation or uniform distribution assumptions.
3.  The proposed methods are supported by both theoretical guarantees and extensive empirical validation.

**Weaknesses:**

1. There is a contradiction between the method's motivation and its empirical results. The second-order strategy (COMES-RL), which was introduced specifically to model label correlations, paradoxically performs significantly _worse_ than the first-order strategy (COMES-HL) on datasets with strong label correlations (e.g., CUB and COCO), an inconsistency the authors do not address.

2. The paper's claim of proposing "unbiased risk estimators" is misleading. The practical estimators (Eq. 8, 14) used in the algorithm are "corrected" versions that are _biased_ (to avoid overfitting from the original unbiased forms).

3. The method's reliance on accurate class-prior estimation ($\pi_j$) is a critical vulnerability.

**Questions:**

See in weekness.

---

> ### Author Response · Authors · 2025-11-25
> **Response to Reviewer 2V1q**
>
> First of all, we are very grateful for your help and time in reviewing our submission. Below are our answers (A) to the weaknesses (W).
>
> **W1: Contradiction between the motivation and its empirical results.**
>
> **A1:** Thanks for the insightful question! Apart from label correlations, loss characteristics can greatly affect model performance. The binary cross-entropy (BCE) loss function is simple yet the most commonly used in multi-label image classification literature due to its excellent loss properties. Even on datasets with strong label correlations, BCE can outperform the ranking loss even though it does not explicitly consider the label ranking relationship. While the ranking loss may be superior in certain cases, such as the noisy versions of VOC2007 and VOC2012 mentioned in the paper, the BCE loss often outperforms it on various multi-label image datasets. We conducted an experiment on CUB and COCO using ordinary labels. We used BCE and ranking loss, respectively. The experimental results are shown below.
>
> |Loss|CUB|COCO2014|
> |---|---|---|
> |BCE|**60.42**|**85.40**|
> |Ranking|60.14|85.03|
>
> As we can see, the BCE loss outperforms the ranking loss on the original CUB and COCO2014 datasets. This demonstrates the superiority of using BCE loss on these datasets. Therefore, it is reasonable that COMES-HL outperforms COMES-RL on the CUB and COCO2014 datasets, as discussed in the paper. In the future, it would be promising to investigate combining the BCE loss's excellent loss properties with consideration of label correlations to improve classification performance.
>
> **W2: The wording "unbiased risk estimators" is not suitable.**
>
> **A2:** We revised "unbiased risk estimators" to "risk estimators." Thank you for your suggestion to make the paper's writing more rigorous.
>
> **W3: The reliance on $\pi\_j$ is a critical vulnerability.**
>
> **A3:** Class prior estimation is an important and classic topic in the literature on weakly supervised learning, including PU learning [1] and noisy label learning [2]. In theory, class priors can be estimated using many off-the-shelf methods [3-6]. Under mild assumptions, it is often proven that the estimated class priors converge to the true class priors. In some real-world applications, we can collect information about class priors. For example, in disease rate prediction, class priors can be calculated based on past statistics. Even if class priors cannot be calculated or identified, we can hire annotators to annotate a small amount of data to estimate them [7] or estimate them based on a small validation set.
>
> To demonstrate the robustness of inaccurately estimated class priors, we conducted further experiments on the mirflickr dataset, shown below. Here, "-E" means that our methods use inaccurately estimated class priors. We found that the performance of our methods with inaccurately estimated class priors is quite strong and comparable to methods with knowledge of class priors. We are conducting more experiments and will include the results when they are finished.
>
> | Approach | Ranking Loss ↓ | One Error ↓ | Hamming Loss ↓ | Average Precision ↑ |
> |---|---|---|---|---|
> | BCE | 0.106 ± 0.008 | 0.275 ± 0.021 | 0.220 ± 0.007 | 0.813 ± 0.011 |
> | CCMN | 0.106 ± 0.011 | 0.282 ± 0.030 | 0.220 ± 0.006 | 0.811 ± 0.016 |
> | GDF | 0.159 ± 0.007 | 0.409 ± 0.027 | 0.277 ± 0.007 | 0.742 ± 0.013 |
> | CTL | 0.130 ± 0.006 | 0.366 ± 0.017 | 0.237 ± 0.006 | 0.772 ± 0.009 |
> | MLCL | 0.498 ± 0.035 | 0.810 ± 0.066 | 0.601 ± 0.020 | 0.446 ± 0.038 |
> | COMES-HL | 0.095 ± 0.009 | 0.171 ± 0.019 | 0.164 ± 0.003 | 0.843 ± 0.013 |
> | COMES-RL | 0.106 ± 0.006 | 0.206 ± 0.036 | 0.186 ± 0.008 | 0.818 ± 0.011 |
> | COMES-HL-E | 0.107 ± 0.008 | 0.133 ± 0.010 | 0.158 ± 0.002 | 0.858 ± 0.007 |
> | COMES-RL-E | 0.104 ± 0.010 | 0.189 ± 0.010 | 0.183 ± 0.006 | 0.824 ± 0.012 |
>
>
> References:
>
> [1] Positive-Unlabeled Learning with Non-Negative Risk Estimator, NeurIPS 2017.
>
> [2] Learning from Corrupted Binary Labels via Class-Probability Estimation, ICML 2015.
>
> [3] Classification with Asymmetric Label Noise: Consistency and Maximal Denoising, COLT 2013.
>
> [4] Mixture Proportion Estimation via Kernel Embedding of Distributions, ICML 2016.
>
> [5] Mixture Proportion Estimation and PU Learning: A Modern Approach, NeurIPS 2021.
>
> [6] Mixture Proportion Estimation Beyond Irreducibility, ICML 2023.
>
> [7] Machine Learning from Weak Supervision: An Empirical Risk Minimization Approach. MIT Press, 2022.

---

> > ### Author Response · Authors · 2025-11-27
> > **Additional Experimental Results Regarding Our Answer A3**
> >
> > We have included additional experimental results regarding A3 on yeastBP, yeastCC, and yeastMF for your reference. Here, "-E" means that our methods use inaccurately estimated class priors. These experimental results demonstrate the superiority of our methods compared with other SOTA methods when class priors are inaccurately estimated.
> >
> > | Approach | One Error ↓-yeastBP | One Error ↓-yeastCC | One Error ↓-yeastMF | Hamming Loss ↓-yeastBP | Hamming Loss ↓-yeastCC | Hamming Loss ↓-yeastMF | Average Precision ↑-yeastBP | Average Precision ↑-yeastCC | Average Precision ↑-yeastMF |
> > |---|---|---|---|---|---|---|---|---|---|
> > | BCE | 0.871 ± 0.008 | 0.814 ± 0.019 | 0.886 ± 0.020 | 0.148 ± 0.007 | 0.162 ± 0.007 | 0.153 ± 0.006 | 0.150 ± 0.013 | 0.487 ± 0.016 | 0.379 ± 0.019 |
> > | CCMN | 0.878 ± 0.016 | 0.823 ± 0.016 | 0.882 ± 0.012 | 0.151 ± 0.007 | 0.163 ± 0.008 | 0.150 ± 0.005 | 0.150 ± 0.012 | 0.479 ± 0.016 | 0.386 ± 0.021 |
> > | GDF | 0.976 ± 0.006 | 0.971 ± 0.008 | 0.972 ± 0.007 | 0.499 ± 0.016 | 0.489 ± 0.026 | 0.497 ± 0.030 | 0.057 ± 0.002 | 0.135 ± 0.010 | 0.144 ± 0.016 |
> > | CTL | 0.970 ± 0.006 | 0.964 ± 0.004 | 0.963 ± 0.010 | 0.493 ± 0.009 | 0.499 ± 0.007 | 0.496 ± 0.006 | 0.060 ± 0.002 | 0.154 ± 0.004 | 0.165 ± 0.013 |
> > | MLCL | 0.961 ± 0.038 | 0.862 ± 0.066 | 0.887 ± 0.066 | 0.881 ± 0.096 | 0.845 ± 0.051 | 0.837 ± 0.024 | 0.082 ± 0.015 | 0.402 ± 0.080 | 0.375 ± 0.124 |
> > | COMES-HL | 0.641 ± 0.030 | 0.744 ± 0.020 | 0.800 ± 0.023 | 0.073 ± 0.008 | 0.119 ± 0.015 | 0.101 ± 0.005 | 0.458 ± 0.020 | 0.657 ± 0.020 | 0.552 ± 0.023 |
> > | COMES-RL | 0.808 ± 0.016 | 0.754 ± 0.022 | 0.805 ± 0.020 | 0.051 ± 0.001 | 0.045 ± 0.004 | 0.048 ± 0.003 | 0.315 ± 0.015 | 0.651 ± 0.023 | 0.549 ± 0.019 |
> > | COMES-HL-E | 0.747 ± 0.020 | 0.803 ± 0.013 | 0.850 ± 0.008 | 0.042 ± 0.003 | 0.082 ± 0.003 | 0.103 ± 0.004 | 0.303 ± 0.008 | 0.475 ± 0.023 | 0.432 ± 0.020 |
> > | COMES-RL-E | 0.957 ± 0.009 | 0.850 ± 0.014 | 0.889 ± 0.006 | 0.051 ± 0.001 | 0.045 ± 0.001 | 0.049 ± 0.001 | 0.106 ± 0.008 | 0.400 ± 0.031 | 0.347 ± 0.009 |

---

### Official Review · Reviewer_fd9d · 2025-10-28

**Soundness:** 4
**Presentation:** 3
**Contribution:** 3
**Rating:** 8
**Confidence:** 5

**Summary:**

This paper proposed the consistent approaches to handle both partial label and complementary multi-label problems in a unified way, with two unbiased risk estimators based on first- and second-order strategies. The theoretical work is elegant and self-contained. In addition, the empirical study largely validates the effectiveness of the proposed approaches.

**Strengths:**

1. The paper innovatively transforms the PML problem into a negative-unlabeled (NU) learning problem through a carefully derived loss function, providing a clear and elegant theoretical perspective.
2. The theoretical analysis is rigorous, with well-stated assumptions, formal consistency proofs, and convergence rate derivations.
3. The empirical evaluation is thorough, covering both standard benchmarks and additional real-world datasets, which enhances the credibility and generality of the proposed method.
4. The paper is well-motivated and clearly written, with smooth logical flow and sound reasoning connecting problem definition, theory, and empirical validation.

**Weaknesses:**

1. The derivation in Lemma 1 heavily relies on assumptions, which may be difficult for readers to intuitively grasp. I would suggest providing more intuitive insights or illustrative examples to clarify the underlying rationale of the lemma and its implications for the overall theoretical framework.
2. The rank loss based on the second-order strategy appears to primarily capture the gap between the ground-truth and non-ground-truth labels, but not the relevance among ground-truth labels themselves. It would be interesting to discuss whether a rank loss can be designed to simultaneously model both the inter-ground-truth relevance and the ground-truth and non-ground-truth gap.
3. In the experiments on synthetic benchmark datasets, the differences between case-a and case-b are not clearly analyzed. It would strengthen the empirical section if the authors could elaborate on the motivation for setting up these two distinct cases, and provide a detailed discussion of the respective findings and their implications.
4. Although this paper focuses on weakly supervised multi-label learning, the proposed loss formulations and theoretical derivations seem readily transferable to weakly supervised multi-class settings, such as partial label learning and complementary label learning. A discussion along with potential preliminary experiments on this broader applicability would further highlight the generality and impact of this paper.

**Questions:**

Please see the weaknesses.

---

> ### Author Response · Authors · 2025-11-25
> **Response to Reviewer fd9d**
>
> Thank you for your great effort in reviewing our paper. We are encouraged that you agree with the contributions of our paper. Below are our answers (A) to the weaknesses (W).
>
> **W1: Lemma 1 is not easy to understand and it is helpful to provide more intuitive insights or illustrative examples.**
>
> **A1:** Lemma 1 indicates a class-wise data generation process of PML. Based on the PML problem definition, the candidate label set for each instance can be regarded as being generated by excluding obviously irrelevant labels. Based on this, we propose the following data generation assumption: We ask annotators to determine whether a label is clearly irrelevant. However, it is difficult to accurately identify all irrelevant labels for a given image, so only some irrelevant labels can be identified. If annotators are uncertain, they should skip this question. We formulate this process as the sampling scheme $p(s_j=0|\boldsymbol{x},y_j=0)=p_j$ in Lemma 1; that is, only some irrelevant labels are considered non-candidate labels. Based on this data generation process, we prove that $p(\boldsymbol{x}|s_j=0)=p(\boldsymbol{x}|y_j=0)$ in Lemma 1. This is the basis for further theoretical derivations. The revised version of the paper provides a discussion and a diagram.
>
> **W2: The development of ranking loss for handling both the inter-ground-truth relevance and the ground-truth and non-ground-truth gap.**
>
> **A2:** Thanks for the insightful question! Yes, handling the inter-ground-truth relevance is also promising. The second-order strategy in this paper is mainly designed for consistently expressing the ranking loss. Another important technique in multi-label classification is label co-occurrence, which may also be helpful for classifier induction [1]. Additionally, determining relevant labels from the candidate label set is important for further exploitation of label co-occurrence. We will consider this in our future work to enhance classification performance.
>
> **W3: The difference between case-a and case-b, and a detailed discussion of the respective findings and their implications, are lacking.**
>
> **A3:** In case-a, irrelevant labels are flipped to candidate labels independently, which is the assumption used in [1]. This strategy is common in learning with noisy labels [2], where PML is a special case of multi-label classification with noisy labels [1]. In case-b, we assign non-candidate labels in a class-wise manner. For each class, we randomly sample a fraction of the training data and assign that class as a non-candidate label. This data generation process corresponds to the assumption proposed in this paper. We use this process to confirm the effectiveness of our proposed method under this assumption. Additionally, since real-world datasets have low noise rates, we selected high flipping rates to evaluate the effectiveness of our proposed methods on challenging datasets with high noise rates. More details can be found in Appendix C.1.
>
> Based on Table 3, we can draw the following conclusions: 1) The proposed COMES-HL and COMES-RL approaches outperform the compared methods in different synthetic dataset cases, thus validating the effectiveness of our approaches in handling various data generation assumptions. 2) CCMN and MLCL are both based on the uniform distribution assumption, which differs from case-a and case-b, which are two more realistic data generation processes. Therefore, they fail to achieve superior performance. 3) Although GDF and CTL use transition matrices to model generation processes, a seemingly more practical assumption, the estimation of generation processes is inaccurate, as discussed in the Introduction section. 4) Our proposed approaches do not rely on these assumptions, and their strong classification performance results from the effectiveness of the proposed risk-correction techniques.
>
> **W4: Future applications to other classification problems.**
>
> **A4:** This paper focuses on multi-label classification problems. Because our proposed framework is very general, developing more methodologies and applying them to additional classification tasks is promising. First, while the paper considers Hamming and ranking losses, developing consistent methods with respect to other multi-label evaluation metrics would be beneficial. Second, identifying suitable scenarios involving complementary or partial labels and applying the developed techniques in the context of multi-class classification problems would be highly beneficial. Thank you for your helpful comments. We will take them into account in our future work.
>
> References:
>
> [1] Counterfactual Reasoning for Multi-Label Image Classification via Patching-Based Training, ICML 2024.

---

### Official Review · Reviewer_Yr8e · 2025-10-29

**Soundness:** 2
**Presentation:** 3
**Contribution:** 3
**Rating:** 4
**Confidence:** 4

**Summary:**

This paper addresses the partial/complementary multi-label learning problem. By proposing a consistent multi-label classification
under inexact supervision framework called COMES, this paper designs two risk-consistent estimators w.r.t. two classic multi-class classification losses, i.e., the Hamming loss and the ranking loss. Compared with previous works, the proposed estimators neither require estimating the generation process of candidate or complementary labels nor rely on the uniform-distribution assumption. In the theoretical parts, this paper derives the generalization bounds for the proposed estimators (COMES-HL and COMES-RL). In experiments, the proposed estimators are evaluated on both real-world and synthetic PML benchmark datasets, achieving lower errors and higher average precision compared with previous methods.

**Strengths:**

- This paper is overall well written and easy to follow.
- The proposed methods are theoretically inspired and proved to be consistent. Although I did not check every detail in the proof, the theoretical results seem sound and reasonable.
- The proposed methods neither require estimating the generation process of candidate or complementary labels nor rely on the uniform-distribution assumption, which is a significant advancement compared to existing methods.

**Weaknesses:**

**Major**
- In Theorem 2, if the non-negative $\alpha=0$, according to (9), the estimator is inconsistent because the bound becomes independent of $n$. For example, if the classification problem is easy enough so that $g_j(x)$ predicts every $y$ exactly, then theoretically, we have $\pi_j\mathbb{E}\_{p(x|y_j=1)} [\ell(g_j(x),1)]=0$. Nonetheless, I believe that the estimator can still be proved consistent in such a corner case, because the classification problem becomes very easy now. I hope the authors could discuss this case in further detail so that the theoretical results can be more rigorous and complete.
- Similarly, in Theorem 5, it is also possible that $\gamma=0$ for easy classification problems, making the bound independent of $n$.
- Another of my major concerns is about the fair comparison in the experiment section. In Figure 2, the inaccurate class priors affect the performance of COMES. Even for the relatively robust COMES-RL, under a slight noise $\theta=0.1$, the average precision drops unacceptably. For example, on mirflickr, mAP drops from 0.818 to approximately 0.80, and on music_style, mAP drops from 0.732 to below 0.70, which makes COMES-RL perform worse than many of the compared methods in Table 2. Therefore, I wonder if these compared methods also use the true class priors in Table 2. If so, from my perspective, an additional comparison under estimated class priors may be more persuasive.

**Minor**
- In Eq.(8), an absolute value function is used to prevent overfitting. Further explanations are needed to elaborate on how and why this approach works.
- In Section 3.1, the generation procedure of non-candidate labels is class-dependent instead of instance-dependent. I think this point should be emphasized in the context.
- Line 250. "$\sup\_{g_i \in \mathcal{G}}\\|g\\|_\infty$". Should it be "$\sup\_{g_i \in \mathcal{G}}\\|g_i\\|\_\infty$"?

**Questions:**

- In Eq. (13), there is no negative loss term to induce overfitting. Why is the flooding regularization technique necessary?
- Why do Eq. (8) and Eq. (14) use different strategies to avoid overfitting?
- Just out of curiosity, why are the proposed methods named first-order and second-order strategies when no derivatives are involved?
- How do the proposed methods perform under estimated class priors?

---

> ### Author Response · Authors · 2025-11-25
> **Response to Reviewer Yr8e (1/2)**
>
> First, we are very grateful for your time and great effort in reviewing this paper. Below are our answers (A) to the weaknesses (W) and questions (Q).
>
> **W1: About the corner case of the non-negative $\alpha$.**
>
> **A1:** Thank you for pointing out this issue. We agree that Theorems 2 and 3 only hold when $\alpha>0$. We have revised the assumption to "We assume that there exists a **positive** constant $\alpha$ such that $\forall j\in\mathcal{Y}, \pi\_j\mathbb{E}\_{p(\boldsymbol{x}|y\_j=1)}\left[\ell\left(g\_j\left(\boldsymbol{x}\right),1\right)\right]\geq\alpha$." This means that, for each class-wise classification risk $\mathbb{E}\_{p(\boldsymbol{x}|y\_j=1)}\left[\ell\left(g\_j\left(\boldsymbol{x}\right),1\right)\right]$, the risk value should be greater than zero. This assumption naturally holds for many loss functions. For example, in COMES-HL, the cross-entropy loss used in the paper cannot become zero due to the assumption about the boundness of the logits: $\sup\_{g\_j\in\mathcal{G}}\\|g\_j\\|\_{\infty} \leq C\_{\mathcal{G}}$.
>
> **W2: About the corner case of the non-negative $\gamma$.**
>
> **A2:** Thanks for pointing out this issue as well. We have revised the assumption as "We assume that there exists a **positive** constant $\gamma$ such that $R_{\mathrm{R}}^{\ell}(\boldsymbol{g})\geq \gamma$." Similar to A1, we assume that the classification risk $R_{\mathrm{R}}^{\ell}(\boldsymbol{g})$ is always positive. This assumption also naturally holds for many symmetric loss functions, such as the sigmoid loss function used in our paper. The value of the sigmoid loss function cannot become zero due to the assumption about the boundness of the logits: $\sup\_{g\_j\in\mathcal{G}}\\|g\_j\\|\_{\infty} \leq C\_{\mathcal{G}}$.
>
> **W3: Robustness of the methods with inaccurate class priors.**
>
> **A3:** Since the class priors are very small, even a small amount of Gaussian noise can greatly impact their value. The purpose of the experiments in Figure 2 of the original version is to demonstrate the robustness of the methods when class priors are highly noisy. In practice, however, we found that class prior estimation methods, such as the method in Appendix A.2, could estimate class priors much more accurately.
>
> To demonstrate the robustness of inaccurately estimated class priors, we conducted further experiments on the mirflickr dataset, shown below. Here, "-E" means that our methods use inaccurately estimated class priors. We found that the performance of our methods with inaccurately estimated class priors is quite strong and comparable to methods with knowledge of class priors. We are conducting more experiments and will include the results when they are finished.
>
> | Approach | Ranking Loss ↓ | One Error ↓ | Hamming Loss ↓ | Average Precision ↑ |
> |---|---|---|---|---|
> | BCE | 0.106 ± 0.008 | 0.275 ± 0.021 | 0.220 ± 0.007 | 0.813 ± 0.011 |
> | CCMN | 0.106 ± 0.011 | 0.282 ± 0.030 | 0.220 ± 0.006 | 0.811 ± 0.016 |
> | GDF | 0.159 ± 0.007 | 0.409 ± 0.027 | 0.277 ± 0.007 | 0.742 ± 0.013 |
> | CTL | 0.130 ± 0.006 | 0.366 ± 0.017 | 0.237 ± 0.006 | 0.772 ± 0.009 |
> | MLCL | 0.498 ± 0.035 | 0.810 ± 0.066 | 0.601 ± 0.020 | 0.446 ± 0.038 |
> | COMES-HL | 0.095 ± 0.009 | 0.171 ± 0.019 | 0.164 ± 0.003 | 0.843 ± 0.013 |
> | COMES-RL | 0.106 ± 0.006 | 0.206 ± 0.036 | 0.186 ± 0.008 | 0.818 ± 0.011 |
> | COMES-HL-E | 0.107 ± 0.008 | 0.133 ± 0.010 | 0.158 ± 0.002 | 0.858 ± 0.007 |
> | COMES-RL-E | 0.104 ± 0.010 | 0.189 ± 0.010 | 0.183 ± 0.006 | 0.824 ± 0.012 |
>
> **W4: About the absolute value function.**
>
> **A4:** There are negative terms in the unbiased risk estimator in Eq. (7). Through empirical analysis, we discovered that directly minimizing Eq. (7) often leads to overfitting issues. Inspired by a commonly used technique in weakly supervised learning literature that employs non-negative risk-correction functions to encapsulate potentially negative terms [1-3], we propose using an absolute value function to encapsulate each term. Using risk-correction functions greatly improves classification performance, as is widely observed in other weakly supervised learning problems [4].
>
> **W5: The generation of candidate labels is instance-independent.**
>
> **A5:** The current literature on partial multi-label learning (PML) and complementary multi-label learning (CML) assumes label generation is instance-independent (see Table 1). Following previous work, we consider the instance-independent case as well. Designing consistent methods for instance-dependent cases is very challenging due to the difficulty of estimating instance-dependent generation processes, as far as we know from the literature on weakly supervised learning [5,6]. In future work, we will consider developing instance-dependent methods with strong theoretical guarantees. We have added the discussion to the paper.

---

> > ### Author Response · Authors · 2025-11-25
> > **Response to Reviewer Yr8e (2/2)**
> >
> > **W6: $\sup\_{g\_j\in\mathcal{G}}\\|g\\|\_{\infty} \leq C\_{\mathcal{G}}$ should be $\sup\_{g\_j\in\mathcal{G}}\\|g\_j\\|\_{\infty}$.**
> >
> > **A6:** Thanks for pointing out the issue. We have revised it in the update version of our paper.
> >
> > **Q1: Why is the flooding regularization technique necessary although there is no negative term in Eq. (13)?**
> >
> > **A7:** Since $\mathbb{E}[R\_{\mathrm{R}}^{\ell}(\boldsymbol{g})] > 0$, we have $\mathbb{E}[\hat{R}\_{\mathrm{R}}^{\ell}(\boldsymbol{g})] > \sum\nolimits\_{1\leq j<k\leq q}Mp(y\_j=0,y\_k=0)$. Therefore, when the value of $\hat{R}\_{\mathrm{R}}^{\ell}(\boldsymbol{g})$ is smaller than $\sum\nolimits\_{1\leq j<k\leq q}Mp(y\_j=0,y\_k=0)$, the original risk estimator can still be negative. This may also cause overfitting problems, so we apply the flooding regularization technique to address these issues.
> >
> > **Q2: Why do Eq. (8) and Eq. (14) use different strategies to avoid overfitting?**
> >
> > **A8:** This is because the value of $\sum\nolimits\_{1\leq j<k\leq q}Mp(y\_j=0,y\_k=0)$ in Eq. (12) is very difficult to estimate. Therefore, we use the flooding technique, which employs a hyperparameter for risk correction. Despite the use of a hyperparameter, we prove that the corrected risk estimator remains consistent under mild assumptions.
> >
> > **Q3: Why call the methods first- and second-order strategies?**
> >
> > **A9:** We use the terminology from [7], where methods that handle different classes independently are called "first-order strategies," and methods that consider pairwise label correlations are called "second-order strategies."
> >
> > **Q4: How do the proposed methods perform under estimated class priors?**
> >
> > **A10:** Please see A3.
> >
> > References:
> >
> > [1] Mitigating Overfitting in Supervised Classification from Two Unlabeled Datasets- A Consistent Risk Correction Approach, AISTATS 2020.
> >
> > [2] Positive-Unlabeled Learning with Non-Negative Risk Estimator, NeurIPS 2017.
> >
> > [3] Binary Classification with Confidence Difference, NeurIPS 2023.
> >
> > [4] Machine Learning from Weak Supervision: An Empirical Risk Minimization Approach. MIT Press, 2022.
> >
> > [5] Identifiability of Label Noise Transition Matrix, ICML 2023.
> >
> > [6] Estimating Instance-dependent Bayes-label Transition Matrix using a Deep Neural Network, ICML 2022.
> >
> > [7] A Review on Multi-Label Learning Algorithms, TKDE 2014.

---

> > > ### Comment · Reviewer_Yr8e · 2025-11-26
> > >
> > > Thank you for the rebuttal. The explanations and additional experimental results addressed my concerns. I decide to raise my recommendation score to 6. I cannot give a higher score because the corner cases are not rigorously proved. (The authors' explanations are also acceptable though, which I think should also be included in the revised manuscript.)

---

> ### Author Response · Authors · 2025-11-27
> **Thanks for your further feedback!**
>
> Thank you very much for your prompt feedback! We are grateful that you raised your recommendation score.
>
> Currently, Theorems 2 and 5 concerning the consistency properties of risk estimators hold with positive $\alpha$ and positive $\gamma$, respectively, due to the use of McDiarmid’s inequality in their proofs. As we explained in our previous response, the corner cases of these assumptions cannot be achieved based on the specific use of logistic and sigmoid losses and the assumption of bounded model outputs. We will consider the corner cases in our future work. We have added a discussion of this issue in Appendix F in the revised version of the manuscript.
>
> Additionally, we have included additional experimental results regarding A3 on yeastBP, yeastCC, and yeastMF for your reference. These results demonstrate the superiority of our methods compared with other SOTA methods when class priors are inaccurately estimated.
>
> | Approach | One Error ↓-yeastBP | One Error ↓-yeastCC | One Error ↓-yeastMF | Hamming Loss ↓-yeastBP | Hamming Loss ↓-yeastCC | Hamming Loss ↓-yeastMF | Average Precision ↑-yeastBP | Average Precision ↑-yeastCC | Average Precision ↑-yeastMF |
> |---|---|---|---|---|---|---|---|---|---|
> | BCE | 0.871 ± 0.008 | 0.814 ± 0.019 | 0.886 ± 0.020 | 0.148 ± 0.007 | 0.162 ± 0.007 | 0.153 ± 0.006 | 0.150 ± 0.013 | 0.487 ± 0.016 | 0.379 ± 0.019 |
> | CCMN | 0.878 ± 0.016 | 0.823 ± 0.016 | 0.882 ± 0.012 | 0.151 ± 0.007 | 0.163 ± 0.008 | 0.150 ± 0.005 | 0.150 ± 0.012 | 0.479 ± 0.016 | 0.386 ± 0.021 |
> | GDF | 0.976 ± 0.006 | 0.971 ± 0.008 | 0.972 ± 0.007 | 0.499 ± 0.016 | 0.489 ± 0.026 | 0.497 ± 0.030 | 0.057 ± 0.002 | 0.135 ± 0.010 | 0.144 ± 0.016 |
> | CTL | 0.970 ± 0.006 | 0.964 ± 0.004 | 0.963 ± 0.010 | 0.493 ± 0.009 | 0.499 ± 0.007 | 0.496 ± 0.006 | 0.060 ± 0.002 | 0.154 ± 0.004 | 0.165 ± 0.013 |
> | MLCL | 0.961 ± 0.038 | 0.862 ± 0.066 | 0.887 ± 0.066 | 0.881 ± 0.096 | 0.845 ± 0.051 | 0.837 ± 0.024 | 0.082 ± 0.015 | 0.402 ± 0.080 | 0.375 ± 0.124 |
> | COMES-HL | 0.641 ± 0.030 | 0.744 ± 0.020 | 0.800 ± 0.023 | 0.073 ± 0.008 | 0.119 ± 0.015 | 0.101 ± 0.005 | 0.458 ± 0.020 | 0.657 ± 0.020 | 0.552 ± 0.023 |
> | COMES-RL | 0.808 ± 0.016 | 0.754 ± 0.022 | 0.805 ± 0.020 | 0.051 ± 0.001 | 0.045 ± 0.004 | 0.048 ± 0.003 | 0.315 ± 0.015 | 0.651 ± 0.023 | 0.549 ± 0.019 |
> | COMES-HL-E | 0.747 ± 0.020 | 0.803 ± 0.013 | 0.850 ± 0.008 | 0.042 ± 0.003 | 0.082 ± 0.003 | 0.103 ± 0.004 | 0.303 ± 0.008 | 0.475 ± 0.023 | 0.432 ± 0.020 |
> | COMES-RL-E | 0.957 ± 0.009 | 0.850 ± 0.014 | 0.889 ± 0.006 | 0.051 ± 0.001 | 0.045 ± 0.001 | 0.049 ± 0.001 | 0.106 ± 0.008 | 0.400 ± 0.031 | 0.347 ± 0.009 |
>
> Thank you again for your help and insightful suggestions, which greatly enhanced this work.

---

### Official Review · Reviewer_oQbS · 2025-10-31

**Soundness:** 3
**Presentation:** 3
**Contribution:** 2
**Rating:** 4
**Confidence:** 3

**Summary:**

The paper proposes the COMES framework for partial/complementary multi‑label learning (as the paper points out, these settings are formally equivalent). The paper focuses on deriving two new unbiased losses for Hamming loss (COMES‑HL) and ranking loss (COMES‑RL) under the setting with the assumption that each label has a different but constant (not instance-dependent) probability of being in the candidate set, while being in fact irrelevant for the sample. The authors prove consistency with finite‑sample bounds for both derived losses and perform experiments on six real‑world PML datasets and four synthetic ones, and compare against 5 different algorithms across Hamming loss, ranking loss, one‑error,  coverage, and AP.

**Strengths:**

- The paper sounds and is easy to read.
- Clear theoretical contribution: unbiased estimators with finite‑sample bounds for both Hamming and ranking loss.
- Useful relaxation of assumptions compared to previous methods.

**Weaknesses:**

- The biggest weakness of experiments is that they assume the priors are known, but the problem of estimating priors is, in this case, complex, and the problem may not be identifiable in some cases.
- It is not clear what dataset is used for Figures 2 and 3; comparison with baselines could also be added there.
- While new losses relax the assumption on uniform distribution, calling them general is an overstatement for me, as obviously, the assumption on constant p_j is still strong and likely untrue in many real-world cases.
- "This data generation process coincides well with the annotation process of candidate labels. For example, when asking annotators to provide candidate labels for an image dataset, we
can show them an image and a class label and ask them to determine whether the image is irrelevant
to that class. This is often an easier question to answer than directly asking all relevant labels, since
it is less demanding to exclude some obviously irrelevant labels." - depends, if there is a lot of labels, is it really better to list irrelevant ones? Not sure about that, but I recall that datasets used in experiments were actually created using crowdsourcing, and candidate label sets were created by taking the union of all assigned label sets. Are there any datasets created in this manner?
- Experiments lack a good description of datasets and baseline methods (also in the appendix).
"We evaluate against five classical baselines commonly used in PML/CML learning." - Only classical? None of it is SOTA? What about the assumption these methods are using, do they also require priors? Also, what worries me is that from those baselines, the simplest BCE performs the best most of the time. This makes me question baseline choice and correctness of presented experiments.
A comment how assumptions of the method match data annotations of benchmark datasets would be nice. Also, are all the real datasets indeed real? I might be wrong, but I think some of them are actually created synthetically (yeast ones?). Limited information on data splits/repetitions etc.
- NIT: There are multiple metrics in MLC called "coverage", I assume the authors use the minimal ranking coverage metric here because the stated lower is better, but it would be nice to have metrics defined in the appendix.
- NIT: It's not clear from the main text what is Case-a and Case-b in Table 3
- NIT: When I use the code link, for every file I select, it says: The requested file is not found. Basically, the code is not accessible at the moment of writing this review.

**Questions:**

Please refer to the weakness

---

> ### Author Response · Authors · 2025-11-25
> **Response to Reviewer oQbS (1/3)**
>
> First, we are very grateful for your time and great effort in reviewing this paper. Below are our answers (A) to the weaknesses (W).
>
> **W1: The weakness of the experiment is the class prior is unknown. The problem is not identifiable sometimes.**
>
> **A1:** Class prior estimation is an important and classic topic in the literature on weakly supervised learning, including PU learning [1] and noisy label learning [2]. In theory, class priors can be estimated using many off-the-shelf methods [3-6]. Under mild assumptions, it is often proven that the estimated class priors converge to the true class priors. Besides, in some real-world applications, we can collect information about class priors. For example, in disease rate prediction, class priors can be calculated based on past statistics. Even if class priors cannot be calculated or identified, we can hire annotators to annotate a small amount of data to estimate them [7] or estimate them based on a small validation set.
>
> To demonstrate the robustness of inaccurately estimated class priors, we conducted further experiments on the mirflickr dataset, shown below. Here, "-E" means that our methods use inaccurately estimated class priors. We found that the performance of our methods with inaccurately estimated class priors is quite strong and comparable to methods with knowledge of class priors. More experimental results can be found in the latest reply.
>
> | Approach | Ranking Loss ↓ | One Error ↓ | Hamming Loss ↓ | Average Precision ↑ |
> |---|---|---|---|---|
> | BCE | 0.106 ± 0.008 | 0.275 ± 0.021 | 0.220 ± 0.007 | 0.813 ± 0.011 |
> | CCMN | 0.106 ± 0.011 | 0.282 ± 0.030 | 0.220 ± 0.006 | 0.811 ± 0.016 |
> | GDF | 0.159 ± 0.007 | 0.409 ± 0.027 | 0.277 ± 0.007 | 0.742 ± 0.013 |
> | CTL | 0.130 ± 0.006 | 0.366 ± 0.017 | 0.237 ± 0.006 | 0.772 ± 0.009 |
> | MLCL | 0.498 ± 0.035 | 0.810 ± 0.066 | 0.601 ± 0.020 | 0.446 ± 0.038 |
> | COMES-HL | 0.095 ± 0.009 | 0.171 ± 0.019 | 0.164 ± 0.003 | 0.843 ± 0.013 |
> | COMES-RL | 0.106 ± 0.006 | 0.206 ± 0.036 | 0.186 ± 0.008 | 0.818 ± 0.011 |
> | COMES-HL-E | 0.107 ± 0.008 | 0.133 ± 0.010 | 0.158 ± 0.002 | 0.858 ± 0.007 |
> | COMES-RL-E | 0.104 ± 0.010 | 0.189 ± 0.010 | 0.183 ± 0.006 | 0.824 ± 0.012 |
>
>
> **W2: Datasets for Figures 2 and 3.**
>
> **A2:** For Figure 2, we used mirflickr, music_emotion, and music_style, as indicated in the legend. We used MirFlickr for Figure 3, as indicated in the figure title. We added them to the main text as well in the revised version.
>
> **W3: Calling the method "general" is an overstatement, and the data generation assumption is untrue in real-world applications.**
>
> **A3:** Here, "general" means that the proposed methodology and data generation process can be tailored to different multi-label classification metrics. In this paper, we used the Hamming loss and the ranking loss, and we plan to extend it to more metrics in the future. For clearer expression, we revised the wording accordingly.
>
> Although the data distribution assumption relies on class-wise sampling with fixed rates, it is more practical than the uniform assumption and it does not require estimation of the data generation process. Thus, the proposed data distribution assumption is more practical than the existing assumptions. Following previous work, this paper also considers the instance-independent case. Designing consistent methods for instance-dependent cases is very challenging due to the difficulty of estimating instance-dependent generation processes, as far as we know from the literature on weakly supervised learning [8,9]. In future work, we will consider developing instance-dependent methods with strong theoretical guarantees.

---

> > ### Author Response · Authors · 2025-11-25
> > **Response to Reviewer oQbS (2/3)**
> >
> > **W4: If there are a lot of labels, is it really better to list irrelevant ones? Are there any datasets created in this manner?**
> >
> > **A4:** It is promising to adopt our label generation scheme when the label space is large, since it is expensive and difficult for annotators to accurately include all the relevant labels when annotating a given image. The obtained labels may contain noise. However, it is cheaper to show annotators a pair of an image and a label and ask if the image is relevant to the label. If not, the label can be considered a complementary label. If the annotator is uncertain about the relevance, the label can be considered a candidate label. It is not necessary to list all the relevant labels. We only need to sample a few and consider the remaining labels as candidate labels associated with this image. Then, we can apply our algorithm. This process is easier and cheaper than asking annotators to provide all relevant labels from a large label space directly.
> >
> > This labeling scheme clearly demonstrates the advantage of our proposed data generation process in reducing labeling costs and efforts for multi-label data. Although classic multi-label datasets are mainly generated using a crowdsourcing strategy, we believe our novel class-wise label generation scheme will reduce the labeling costs of multi-label data in the future.
> >
> > **W5: Some details about the experimental setup.**
> >
> > **A5:** **About compared methods:** In this paper, we focus primarily on consistent approaches for PML and CML. We have included all existing consistent approaches from the literature, most of which are up-to-date and state-of-the-art. Our proposed loss function can serve as a base loss and be combined with advanced techniques to improve performance.
> >
> > **About the assumptions and performance of compared methods:** Both CCMN and MLCL are based on the uniform distribution assumption. This differs from the data generation processes of real-world datasets, case-a, and case-b. Therefore, they fail to achieve satisfactory performance. Although GDF and CTL use transition matrices to model generation processes, a more practical assumption that matches case-a, estimation of generation processes is inaccurate, as discussed in the Introduction section. Additionally, they experience significant overfitting issues when handling weak supervision. Our proposed risk-correction techniques mitigate these issues, leading to better performance. We have added a discussion to Appendix E.
> >
> > **About real-world datasets:** Here, mirflickr, music\_emotion, music\_style, yeastBP, yeastCC, and yeastMF are all real-world datasets (please see Section 4.1.1 in [10] for details of mirflickr, music\_emotion, and music\_style). For yeastBP, yeastCC, and yeastMF, we cited the data generation processes from Section IV.A in [11].
> >
> > > The other three datasets, YeastBP,YeastCC, and YeastMF, are protein-protein interaction datasets collected from BioGrid3. We downloaded the functional annotations of Yeast proteins archived on different periods (historical: 2016-03-14, recent: 2017-03-13) from the Gene Ontology, and took the annotations available in history but absent in more recent times as noisy labels. Functional labels of proteins are divided in three orthogonal branches of the Gene Ontology: cellular component (CC), molecular function (MF), and biological process (BP). These functional labels are rather unbalanced. Many labels are associated to no more than 30 proteins, and few labels are associated to more than 300 proteins. To mitigate the imbalance impact, we consider labels that are associated to at least 100 proteins and at most 300 proteins for the experiments. As a result, we consider 50 CC labels, 39 MF labels, and 217 BP labels for YeastCC, YeastMF, and YeastBP, respectively. The numbers of noisy annotations of these three datasets are 260, 234, and 2385,respectively.
> >
> > Since the irrelevant labels in the candidate label set are "annotations available in the past but not in recent times," we believe that they should be considered real annotations.
> >
> > **About data split:** We performed ten-fold cross-validation on real-world datasets. This means that we used nine folds for training and one fold for testing. Then, we recorded the mean accuracy and standard deviation. For the synthetic datasets, we generated synthetic labels three times and recorded the mean accuracy and standard deviation. Finally, we conducted paired t-tests at a 0.05 significance level.

---

> > > ### Author Response · Authors · 2025-11-25
> > > **Response to Reviewer oQbS (3/3)**
> > >
> > > **W6: Definitions of metrics.**
> > >
> > > **A6:** We have added the definitions of evaluation metrics in Appendix C.4.
> > >
> > > **W7: Case-a and Case-b.**
> > >
> > > **A7:** In case-a, irrelevant labels are flipped to candidate labels independently, which is the assumption used in [12]. This strategy is common in learning with noisy labels [13], where PML is a special case of multi-label classification with noisy labels [12]. In case-b, we assign non-candidate labels in a class-wise manner. For each class, we randomly sample a fraction of the training data and assign that class as a non-candidate label. This data generation process corresponds to the assumption proposed in this paper. We use this process to confirm the effectiveness of our proposed method under this assumption. Additionally, since real-world datasets have low noise rates, we selected high flipping rates to evaluate the effectiveness of our proposed methods on challenging datasets with high noise rates. More details can be found in Appendix C.1.
> > >
> > > **W8: About the code link.**
> > >
> > > **A8:** We checked the anonymous link and found that it worked well. For your convenience, we created another anonymous link: https://github.com/ICLR2026-10534/COMES.
> > >
> > >
> > > References:
> > >
> > > [1] Positive-Unlabeled Learning with Non-Negative Risk Estimator, NeurIPS 2017.
> > >
> > > [2] Learning from Corrupted Binary Labels via Class-Probability Estimation, ICML 2015.
> > >
> > > [3] Classification with Asymmetric Label Noise: Consistency and Maximal Denoising, COLT 2013.
> > >
> > > [4] Mixture Proportion Estimation via Kernel Embedding of Distributions, ICML 2016.
> > >
> > > [5] Mixture Proportion Estimation and PU Learning: A Modern Approach, NeurIPS 2021.
> > >
> > > [6] Mixture Proportion Estimation Beyond Irreducibility, ICML 2023.
> > >
> > > [7] Machine Learning from Weak Supervision: An Empirical Risk Minimization Approach. MIT Press, 2022.
> > >
> > > [8] Identifiability of Label Noise Transition Matrix, ICML 2023.
> > >
> > > [9] Estimating Instance-dependent Bayes-label Transition Matrix using a Deep Neural Network, ICML 2022.
> > >
> > > [10] Partial Multi-Label Learning via Credible Label Elicitation, TPAMI 2021.
> > >
> > > [11] Feature-Induced Partial Multi-label Learning, ICDM 2018.
> > >
> > > [12] CCMN: A general framework for learning with class-conditional multi-label noise, TPAMI 2023.
> > >
> > > [13] Co-teaching: Robust training of deep neural networks with extremely noisy labels, NeurIPS 2018.

---

> > > > ### Author Response · Authors · 2025-11-27
> > > > **Additional Experimental Results Regarding Our Answer A1**
> > > >
> > > > We have included additional experimental results regarding A1 on yeastBP, yeastCC, and yeastMF for your reference. Here, "-E" means that our methods use inaccurately estimated class priors. These experimental results demonstrate the superiority of our methods compared with other SOTA methods when class priors are inaccurately estimated.
> > > >
> > > > | Approach | One Error ↓-yeastBP | One Error ↓-yeastCC | One Error ↓-yeastMF | Hamming Loss ↓-yeastBP | Hamming Loss ↓-yeastCC | Hamming Loss ↓-yeastMF | Average Precision ↑-yeastBP | Average Precision ↑-yeastCC | Average Precision ↑-yeastMF |
> > > > |---|---|---|---|---|---|---|---|---|---|
> > > > | BCE | 0.871 ± 0.008 | 0.814 ± 0.019 | 0.886 ± 0.020 | 0.148 ± 0.007 | 0.162 ± 0.007 | 0.153 ± 0.006 | 0.150 ± 0.013 | 0.487 ± 0.016 | 0.379 ± 0.019 |
> > > > | CCMN | 0.878 ± 0.016 | 0.823 ± 0.016 | 0.882 ± 0.012 | 0.151 ± 0.007 | 0.163 ± 0.008 | 0.150 ± 0.005 | 0.150 ± 0.012 | 0.479 ± 0.016 | 0.386 ± 0.021 |
> > > > | GDF | 0.976 ± 0.006 | 0.971 ± 0.008 | 0.972 ± 0.007 | 0.499 ± 0.016 | 0.489 ± 0.026 | 0.497 ± 0.030 | 0.057 ± 0.002 | 0.135 ± 0.010 | 0.144 ± 0.016 |
> > > > | CTL | 0.970 ± 0.006 | 0.964 ± 0.004 | 0.963 ± 0.010 | 0.493 ± 0.009 | 0.499 ± 0.007 | 0.496 ± 0.006 | 0.060 ± 0.002 | 0.154 ± 0.004 | 0.165 ± 0.013 |
> > > > | MLCL | 0.961 ± 0.038 | 0.862 ± 0.066 | 0.887 ± 0.066 | 0.881 ± 0.096 | 0.845 ± 0.051 | 0.837 ± 0.024 | 0.082 ± 0.015 | 0.402 ± 0.080 | 0.375 ± 0.124 |
> > > > | COMES-HL | 0.641 ± 0.030 | 0.744 ± 0.020 | 0.800 ± 0.023 | 0.073 ± 0.008 | 0.119 ± 0.015 | 0.101 ± 0.005 | 0.458 ± 0.020 | 0.657 ± 0.020 | 0.552 ± 0.023 |
> > > > | COMES-RL | 0.808 ± 0.016 | 0.754 ± 0.022 | 0.805 ± 0.020 | 0.051 ± 0.001 | 0.045 ± 0.004 | 0.048 ± 0.003 | 0.315 ± 0.015 | 0.651 ± 0.023 | 0.549 ± 0.019 |
> > > > | COMES-HL-E | 0.747 ± 0.020 | 0.803 ± 0.013 | 0.850 ± 0.008 | 0.042 ± 0.003 | 0.082 ± 0.003 | 0.103 ± 0.004 | 0.303 ± 0.008 | 0.475 ± 0.023 | 0.432 ± 0.020 |
> > > > | COMES-RL-E | 0.957 ± 0.009 | 0.850 ± 0.014 | 0.889 ± 0.006 | 0.051 ± 0.001 | 0.045 ± 0.001 | 0.049 ± 0.001 | 0.106 ± 0.008 | 0.400 ± 0.031 | 0.347 ± 0.009 |

---

### Official Review · Reviewer_aB8Q · 2025-11-01

**Soundness:** 3
**Presentation:** 3
**Contribution:** 3
**Rating:** 8
**Confidence:** 3

**Summary:**

The paper addressed the Partial and Complementary multi-label learning in a unified way. The authors proposed unbiased risk estimators for hamming and ranking loss under inexact supervision, and provided consistency convergence guarantees. In comparison to existing approaches, the proposed method does not rely on accurate estimation of data generation process. They also showed the effectiveness of the framework through empirical results.

**Strengths:**

Originality: The problem of PML and CML have been addressed separately but this paper provides a unified framework to address the two settings. The authors proposed the method which generalize the label generation process (instead of naively assuming the process) for risk estimators.

Quality: The methodology demonstrates technical soundness. The assumptions and lemmas have extensive proofs and details. Based on the experimental section, the method seems to perform well empirically too.

Clarity: The paper is mainly easy to follow and has proper motivation of the problem setting.

Significance: The paper makes substantial contributions to the area of multi-label learning. I think the results are significant (the theoretical and experimental, both).

**Weaknesses:**

1. Although the method is more general than prior work, it depends on a strong assumption used in Lemma 1, which results in the independence of the data generation process from the samples. (Authors mentioned this in the paper as well)
2. I am particularly focused on the result of the Lemma that $p(x|s_j = 0) = p(x|y_j = 0)$. This assumption seems to be a very strong signal for negative labels. This implies a perfect annotation process for 'irrelevant' labels. The proof of the lemma uses another assumption: $p(j \notin Y | x, s_j = 0) =p(j \notin Y | s_j = 0). $, which has not been explained in the paper.
3. The experimental section could be improved a lot. The authors have done experiments but the rationale behind case A and case B is not mentioned. Why choose this particular method of generating the labels and why not some other method (or different \tau)?
4. For the effectiveness of the framework, and the impact of the data generation process, there is a need for more experiments. What would be the effect of changing the dataset size for training?

**Questions:**

1. In figure 2, the experiments are done with different sigmas for class priors. Why not actually modify the flip rate in case A for example, and then evaluate?
2. The average precision of mirflickr stays the same as you increase sigma. Why?
3. This paper consider data generation process independent from the samples. Shouldn’t MLCL perform better because it is estimating the data generation process from the samples? Can you elaborate? Because I would assume that estimating the data generation process based on the knowledge of the actual samples should be better than independent data generation process (in this case, flipping the negative labels to positive labels with prob 0.9).
4. Theorem 4 assumes that $l$ is symmetric but in reality, the used loss binary cross entropy, is not symmetric. Am I missing something?

---

> ### Author Response · Authors · 2025-11-25
> **Response to Reviewer aB8Q (1/2)**
>
> First, we would like to express our gratitude for your time and efforts in reviewing our paper. We are encouraged and thankful that you agree with the paper's contributions. Below are our answers (A) to the weaknesses (W) and questions (Q).
>
> **W1: The method depends on a strong assumption in Lemma 1 that results in the independence of the data generation process from the samples.**
>
> **A1:** The current literature on partial multi-label learning (PML) and complementary multi-label learning (CML) assumes label generation is instance-independent (see Table 1). Following previous work, we consider the instance-independent case as well. Designing consistent methods for instance-dependent cases is very challenging due to the difficulty of estimating instance-dependent generation processes, as far as we know from the literature on weakly supervised learning [3,4]. In future work, we will consider developing instance-dependent methods with strong theoretical guarantees.
>
> **W2: Lemma 1 seems to be a very strong signal for negative labels. Besides, the equation $p(j\notin Y|\boldsymbol{x}, s_{j}=0)=p(j\notin Y|s_{j}=0)=1$ has not been explained in the paper.**
>
> **A2:** Because the original problem assumptions of PML and CML are very strong, Lemma 1 is naturally strong. PML assumes that all relevant labels are included in the candidate label set. Conversely, CML assumes that all complementary labels are perfectly irrelevant. Since this paper works with standard PML and CML settings, our assumptions are naturally strong to accommodate the strict characteristics of the problem settings of PML and CML. It is promising to explore new settings of imperfect PML or CML and develop methodologies accordingly in the future.
>
> We have included more detailed derivations of the proof of Lemma 1 in the revised version. According to the definition of PML, when $s_{j}=0$, the $j$-th class is unlikely to be a relevant label, so we have $p(j\notin Y|\boldsymbol{x}, s_{j}=0)=p(j\notin Y|s_{j}=0)=1$.
>
> **W3: More descriptions of the data generation processes in the experiments.**
>
> **A3:** For synthetic datasets, we consider two data generation processes. In case-a, irrelevant labels are flipped to candidate labels independently, which is the assumption used in [1]. This strategy is common in learning with noisy labels [2], where PML is a special case of multi-label classification with noisy labels [1]. In case-b, we assign non-candidate labels in a class-wise manner. For each class, we randomly sample a fraction of the training data and assign that class as a non-candidate label. This data generation process corresponds to the assumption proposed in this paper. We use this process to confirm the effectiveness of our proposed method under this assumption. Additionally, since real-world datasets have low noise rates, we selected high flipping rates to evaluate the effectiveness of our proposed methods on challenging datasets with high noise rates. More details can be found in Appendix C.1.
>
> **W4: What effect would changing the dataset size for training have?**
>
> **A4:** The datasets span a broad range of sizes, from 6,139 (yeastBP) to 123,287 (COCO2014). We believe that conducting experiments on such diverse datasets of various sizes confirms the superiority of our proposed methods. Classification performance increases with more training data for the same dataset. This phenomenon is also explained by Theorems 3 and 6, which state that the estimation error bound decreases with more training data.
>
> **Q1: Why not adjust the flipping rate rather than the class priors in Figure 2?**
>
> **A5:** Since the class priors are parameters of the proposed methods, we evaluated the effect of changing their values on model performance. However, this cannot be achieved by modifying the flipping rates, which control the number of candidate labels in the synthetic datasets.
>
> **Q2: Why does the average precision of mirflickr stay the same as you increase $\sigma$?**
>
> **A6:** This indicates that the average precision of the proposed methods remains consistent within certain ranges of $\sigma$.
>
> **Q3: Why doesn't MLCL perform better even though it estimates the data generation processes?**
>
> **A7:** As discussed in the Introduction section, accurately estimating the data generation process is very difficult. During implementation, we discovered that the estimated noise rates for MLCL differed significantly from the actual values, negatively impacting its performance.

---

> ### Author Response · Authors · 2025-11-25
> **Response to Reviewer aB8Q (2/2)**
>
> **Q4: Theorem 4 uses a symmetric loss, but use the cross-entropy loss.**
>
> **A8:** Thank you for pointing out this issue in the writing. In our implementation, we instantiated $\ell$ with binary cross-entropy loss for COMES-HL and sigmoid loss for COMES-RL. We apologize for the incomplete expression in the original version of the paper and have corrected it in the revised version.
>
>
> References:
>
> [1] CCMN: A general framework for learning with class-conditional multi-label noise, TPAMI 2023.
>
> [2] Co-teaching: Robust training of deep neural networks with extremely noisy labels, NeurIPS 2018.
>
> [3] Identifiability of Label Noise Transition Matrix, ICML 2023.
>
> [4] Estimating Instance-dependent Bayes-label Transition Matrix using a Deep Neural Network, ICML 2022.

---

### Official Review · Reviewer_bJe9 · 2025-11-01

**Soundness:** 3
**Presentation:** 3
**Contribution:** 3
**Rating:** 6
**Confidence:** 4

**Summary:**

This paper addresses multi-label classification under inexact supervision task, proposing a unified framework COMES for partial multi-label learning (PML) and complementary multi-label learning (CML). Existing methods require estimating label generation processes or rely on uniform distribution assumptions. The paper assumes that candidate labels are generated by querying whether an instance is irrelevant to each class, and designs first-order and second-order unbiased risk estimators. The first-order strategy decomposes the problem into multiple binary classification problems using Hamming loss, while the second-order strategy takes label correlations into account using ranking loss. Theoretically, the paper proves consistency and derives convergence rates for both Hamming and ranking losses, improving generalization through absolute value wrapping and flooding regularization. Experiments validate the effectiveness on six real-world and four synthetic datasets, with COMES significantly outperforming baselines including CCMN, GDF, CTL, and MLCL in most cases.

**Strengths:**

1.	The unified perspective on treating PML and CML as equivalent problems is interesting.
2.	This paper not only proves consistency of risk estimators but also derives convergence rates for estimation errors. It establishes generalization bounds by Rademacher complexity analysis, and proves that minimizing the corrected risk estimator can achieve Bayes risk.
3.	This paper covers different types of datasets (images, audio, biological information, etc.) with reasonable settings. Sensitivity analyses evaluate the impact of inaccurate class priors and hyperparameter β, and ablation studies validate the necessity of each module.

**Weaknesses:**

1.	Algorithm 1 contains an obvious error in line 8 of the pseudocode. The conditional branch "else if using the COMES-HL algorithm then" should be changed to COMES-RL rather than COMES-HL.
2.	This paper provides a detailed introduction to first-order and second-order strategies and validates their effectiveness through experiments. However, it does not explain how to reasonably select or combine first-order and second-order strategies in practical applications.

**Questions:**

1.	Is the conditional branch error in Algorithm 1, line 8 a typesetting issue or an actual error in the code? Does this error exist in the algorithm implementation, or is it a description error in the main text?
2.	What is the reason for using different network architectures (MLP and ResNet-50) in experiments? Does this design affect the fairness and comparability of experimental results?
3.	How should first-order and second-order strategies be chosen in practical applications? Has the author considered designing a mechanism to automatically select one strategy or a weighted combination of the two strategies based on dataset characteristics, such as label correlation strength or dataset size?

---

> ### Author Response · Authors · 2025-11-25
> **Response to Reviewer bJe9**
>
> Thank you for taking the time to review our paper. We are very grateful for your helpful and high-quality comments and suggestions. Below are our answers (A) to the weaknesses (W) and questions (Q).
>
> **W1: Typo in line 8 of Algorithm 1's pseudocode.**
>
> **A1:** Thank you for pointing out the typo. Line 8 of Algorithm 1 should read, "else if using the COMES-RL algorithm then." We apologize for the error and have corrected it in the revised version.
>
> **W2: It does not explain how to select or combine first-order and second-order strategies.**
>
> **A2:** The selection of first- and second-order strategies is an important and classic problem in the multi-label classification (MLC) literature [1,2]. Using a second-order strategy is not always better because overemphasizing label co-occurrence can sometimes misguide model training and lead to a degradation in classification performance [2,3]. Therefore, the choice depends on the specific classification task, data properties, and practical performance. Our two proposed methods are complementary and compatible with different datasets to achieve satisfactory performance.
>
> **Q1: Does the error exist in the algorithm implementation, or is it a description error in the main text?**
>
> **A3:** This is only a typo, and our algorithm implementation is correct.
>
> **Q2: What is the reason for using different network architectures (MLP and ResNet-50) in experiments?**
>
> **A4:** It is because the six real-world datasets are tabular data, whereas the other four synthetic datasets are image data. Therefore, we used an MLP for the six tabular datasets. Following widely used protocols [4], we used a pre-trained ResNet for image datasets. Notably, the comparison is fair since we used the same network architecture for the methods being compared.
>
> **Q3: Has the author considered designing a mechanism to automatically select one strategy or a weighted combination of the two strategies based on dataset characteristics, such as label correlation strength or dataset size?**
>
> **A5:** Our proposed first- and second-order strategies complement each other and can be chosen on a case-by-case basis depending on the properties of the specific dataset and practical performance. We will consider investigating the optimal ensemble strategy in future work.
>
> References:
>
> [1] A Review on Multi-Label Learning Algorithms, TKDE 2014.
>
> [2] Counterfactual Reasoning for Multi-Label Image Classification via Patching-Based Training, ICML 2024.
>
> [3] Boosting Multi-Label Image Classification with Complementary Parallel Self-Distillation, IJCAI 2022.
>
> [4] Multi-Label Learning from Single Positive Labels, CVPR 2021.

---

> > ### Comment · Reviewer_bJe9 · 2025-11-28
> >
> > The authors have addressed my concern, so I will raise my score.

---

### Author Response · Authors · 2025-12-01
**Summary of the Rebuttal**

Dear SACs, ACs, and reviewers,

Thank you for taking the time to review our submission. Our paper has benefited greatly from your suggestions and comments.

During the rebuttal process, two of the six reviewers replied to our rebuttal. We were pleased to see that Reviewer Yr8e believed our response addressed their issues, **raising their score from 4 to 6**. Reviewer bJe9 also said they would **raise the score** because we addressed their concerns. Additionally, although Reviewer oQbS did not respond during the rebuttal process, we believe our additional clarifications regarding class prior estimation, experimental details, and supplementary experiments on the robustness of our methods sufficiently address their concerns.

We are grateful that several of our strengths and contributions were recognized, including the following:
- **Interesting unified perspective on treating partial multi-label learning (PML) and complementary multi-label learning (CML) as equivalent problems** (*Reviewers bJe9, aB8Q, and 2V1q*).
- **Solid theoretical guarantees and contributions** (*Reviewers bJe9, aB8Q, oQbS, Yr8e, fd9d, and 2V1q*): Our proposed approaches are supported by sufficient theoretical guarantees, including well-stated assumptions, formal consistency proofs, and convergence rate derivations.
- **Novel and practical proposals about the data generation processes** (*Reviewers aB8Q, oQbS, and Yr8e*): The proposed data generation processes generalize the uniform distribution and do not require estimation of the generation process.
- **Clear motivation and writing** (*Reviewers aB8Q, oQbS, Yr8e, fd9d, and 2V1q*).
- **Extensive experimental results** (*Reviewers bJe9, aB8Q, fd9d, and 2V1q*): The extensive experimental results clearly validate the effectiveness of the proposed methods.

In our rebuttal, we addressed each of the reviewers' concerns point by point and revised the manuscript accordingly. The major points are as follows:

- **Class prior estimation**: In the rebuttal, we added more experimental results showing that the proposed methods are robust to inaccurately estimated class priors. Additionally, we respectfully point out that class priors can be estimated with theoretical guarantees or obtained in many ways.
- **More details of experiments**: We have added more details about the experimental setup and analyses in Appendices C and E.
- **Limitation**: We acknowledge that we followed previous work in considering instance-independent cases, and we plan to address instance-dependent cases in future work.
- **Corner cases of assumptions and typos**: We have updated the discussion of the corner cases of the assumptions and corrected some typos.

Thank you again for your help and time in reviewing this submission!

Best regards,

Authors of ICLR2026 Submission 10534

---

### Meta-Review · Area_Chair_9AX6 · 2026-01-21

**Summary:**

This paper has received mixed reviews 6, 8, 4, 4, 8 ,6. The paper studies MML under first-order and second-order (correlation capturing) MLC. The authors unify PML and CML and provide theoretical guarantees. Experiments on tabular and image datasets are provided. Reviewers have raised few concerns particularly about strong assumption in Lemma 1, assumption on "known priors", alpha=0 case in Theorem 2. On balance, authors addressed these concerns and supplemented additional experiments. The authors are urged to prepare a thorough revision of this paper for camera-ready.

**Reviewer Scores:**

I believe Reviewer oQbS and Reviewer Yr8e would raise their scores to 6 as their key concerns regarding prior and theorem 2 appear resolved.

---

### Decision · Program_Chairs · 2026-01-26

Accept (Poster)